# Comparison of Two Distinct Subpopulations of *Klebsiella pneumoniae* ST16 Co-Occurring in a Single Patient

Biying Zhang,[a] Renjing Hu,[b] Qinghua Liang,[a] Shuang Liang,[a] Qin Li,[a] Jiawei Bai,[a] Manlin Ding,[a] Feiyang Zhang,[a] Yingshun Zhou[a]

aDepartment of Pathogenic Biology, School of Basic Medicine, Southwest Medical University, Luzhou, Sichuan, China
bDepartment of Laboratory Medicine, The Affiliated Wuxi No.2 People's Hospital of Nanjing Medical University, Wuxi, Jiangsu, China

Biying Zhang and Renjing Hu contributed equally to this article. Author order was determined based on seniority.

**ABSTRACT** The higher resistance rate to ceftazidime-avibactam (CZA) is mainly related to carbapenem resistance, especially New Delhi metallo-$\beta$-lactamase (NDM). The CZA-susceptible *Klebsiella pneumoniae* (K191663) and the later CZA-resistant isolates (K191724, K191725, K191773) co-producing NDM-4 and OXA-181 were obtained from the same hospitalized patient returning from Vietnam. Our study aims to elucidate the diversity of *K. pneumoniae* ST16 through comparative analysis of whole-genome sequencing (WGS) data and identify the potential evolution of plasmids by sequencing longitudinal clinical isolates during antibiotic treatment. Firstly, multilocus sequence typing analysis and phylogenic analysis suggested that these strains belong to the two lineages of *K. pneumoniae* ST16. Surprisingly, the CZA-resistant strains were closely related to *K. pneumoniae* ST16 described in South Korea, instead of the $bla_{NDM-4}$- or $bla_{OXA-181}$-carrying ST16 reported in Vietnam. Secondly, $bla_{NDM-4}$, $bla_{TEM-1B}$, and *rmtB* co-existed on a self-conjugative IncFII(Yp)-like plasmid, which played a significant role in CZA resistance. It could transfer into the recipient *Escherichia coli* J53 at high frequency, indicating the risk of mobile carbapenemases. In addition, the loss of 12-kbp fragment occurred in $bla_{NDM-4}$-positive isolate (K191773), which was likely caused by insertion sequence-mediated homologous recombination. Last but not least, as a repressor of *acrAB* operon system, *acrR* was truncated by a frameshift mutation in K191663. Thus, our study provided baseline information for monitoring the occurrence and development of bacterial resistance.

**IMPORTANCE** As a leading health care-acquired infection pathogen, *Klebsiella pneumoniae* is threatening a large number of inpatients due to its diverse antibiotic resistance and virulence factors. Heretofore, with a growing number of reports about the coexistence of several carbapenemases in carbapenem-resistant *K. pneumoniae* (CRKP), epidemiologic surveillance has been strengthened. Nevertheless, the nosocomial outbreaks by CRKP ST16 are gradually increasing worldwide. Our study provides a deeper insight into the diversification of clinical isolates of CRKP ST16 in China. In addition, the comparison analysis of resistant plasmids may reveal the transmission of carbapenemase-encoding genes. Furthermore, our study also highlights the importance of longitudinal specimen collection and continuous monitoring during the treatment, which play a crucial role in understanding the development of antibiotic resistance and the evolution of resistance plasmids.

**KEYWORDS** carbapenemase-producing *Klebsiella pneumoniae*, ST16, NDM-4, OXA-181, ceftazidime–avibactam, IS26, whole-genome sequencing

Address correspondence to Yingshun Zhou, yingshunzhou@swmu.edu.cn.

The authors declare no conflict of interest.

The rapid spread of carbapenemase-producing *Klebsiella pneumoniae* poses a serious global threat to patients and public health (1). As a combination of a broad-spectrum cephalosporin with a non-$\beta$-lactam $\beta$-lactamase inhibitor, ceftazidime-avibactam (CZA)

**TABLE 1** Antibiotic susceptibilities of four isolates and their transconjugants (mg/L)[a]

| Isolate | MIC (mg/L) | | | | | | | | | | | | |
|---|---|---|---|---|---|---|---|---|---|---|---|---|---|
| | AMP | AMK | CEF | CAZ | CZA | CTX | IPM | MEM | PB | CIP | TGC | TET | C |
| K191663 | 1024 | 64 | 1024 | 128 | **2/4** | >128 | 8 | 32 | **0.5** | >128 | 2 | 256 | **4** |
| K191724 | >1024 | 1024 | >1024 | >128 | >256/4 | >128 | >128 | >128 | **0.5** | >128 | 1 | **4** | <u>16</u> |
| K191725 | >1024 | 512 | >1024 | >128 | >256/4 | >128 | >128 | >128 | **0.5** | >128 | 1 | **4** | <u>16</u> |
| K191773 | >1024 | 1024 | >1024 | >128 | >256/4 | >128 | >128 | >128 | **0.5** | >128 | 1 | **4** | 8 |
| *E. coli* J53 | **8** | **8** | **4** | **0.5** | **1/4** | **0.125** | **0.25** | **0.125** | 0.5 | **0.125** | **0.25** | **1** | **4** |
| Conjugant J63-OXA1 | 512 | 64 | 512 | 16 | **1/4** | >128 | **0.25** | **0.125** | 0.5 | <u>0.5</u> | **0.5** | 128 | **4** |
| Conjugant J73-NDM4 | >1024 | >1024 | >1024 | >128 | >256/4 | >128 | >128 | 128 | **0.5** | <u>0.5</u> | **0.5** | 2 | **4** |
| *E. coli* ATCC 25922 | **8** | **2** | **8** | **0.125** | | **0.12** | **0.12** | **0.016** | **1** | **0.016** | **0.25** | **1** | 8 |
| *K. pneumoniae* ATCC 700603 | | | | | **1/4** | | | | **0.5** | | | | |

[a]AMP, ampicillin; AMK, amikacin; CEF, cefuroxime sodium; CAZ, ceftazidime; CZA, ceftazidime-avibactam; CTX, cefotaxime; IPM, imipenem; MEM, meropenem; PB, polymyxin B; CIP, ciprofloxacin; TGC, tigecycline; TET, tetracycline; C, chloramphenicol. Bold text indicates susceptible and underlined text represents intermediate.

shows great activity against KPC-, AmpC-, and some OXA48-like-producing strains. However, it is ineffective against metallo-$\beta$-lactamases (MBL), such as New Delhi metallo-$\beta$-lactamase (NDM) (2). NDM-like genes were most widely distributed in Asia, and the primary type is $bla_{NDM-1}$ in China (3). Of the NDM-1 variants, NDM-4 was first detected in India in 2010 (4). Since then, NDM-4-bearing *Enterobacteriaceae* has been increasingly reported worldwide, including Italy (5), Egypt (6), and South Korea (7), while shown sporadic spread in China (8, 9). At the same time, OXA-181, sharing the similar penicillin and carbapenem hydrolytic activity with OXA-48, was associated with nosocomial outbreak by conjugative plasmid IncX3 in Ghana recently (10). The coexistence of two classes of carbapenemases has been detected globally, which could intensify substantial challenges for treatment (6, 7, 11). In this study, we characterized two distinct lineages of *K. pneumoniae* ST16 from the same patient. In 2017 and 2021, *K. pneumoniae* ST16 harboring $bla_{NDM-4}$ or $bla_{OXA-181}$ were reported in Vietnam successively (12, 13). Although ST11 is the major clone of carbapenem-resistant *K. pneumoniae* (CRKP) in China (14), recent reports of ST16 are noteworthy, especially as it was related to hospital-acquired infections (15).

Pathogenic bacteria face challenging new selection pressures when infecting a new host. The selective pressure exerted by the antibiotics has driven point mutations in bacteria and acquiring resistance genes through horizontal gene transfer (16). After developing antibiotic resistance (AR), bacteria may persist and play a key role in recurrent infection after a successful course of therapy with antibiotics (17). The bacterial insertion sequences (ISs) strongly correlate with the evolution of resistance plasmids (18). Belonging to the IS*6* family, IS*26* was the best-studied member and has been associated with gene amplification resulting in AR in a clinical isolate (19). Nevertheless, as bacteria acquire new genes, there are also fitness costs due to gene expression or protein synthesis, which may facilitate the excision of unstable elements (20).

Outer membrane porins (OmpK35 and OmpK36) and AcrAB-TolC efflux pump are known to be the intrinsic mechanisms of multi-drug resistance in Gram-negative bacteria (21, 22). Mutations in *acrR*, a repressor of the *acrAB* operon system, can confer clarithromycin resistance by increasing the transcription of *acrB* in *Haemophilus influenzae* (23). It has been reported that the virulence and AR of *K. pneumoniae* ST258 were increased through adaptive mutations during the colonization *in vivo* (24). Similarly, genome diversity analysis of clinical *K. pneumoniae* strains collected longitudinally from the same patient might contribute to controlling the long-term carrying and dissemination of CRKP.

## RESULTS

**Resistance phenotypes and genomic characteristics of all isolates.** MICs of 4 carbapenem-resistant isolates were listed in Table 1. All isolates expressed resistance to ampicillin, amikacin, cefuroxime sodium, ceftazidime, cefotaxime, imipenem, meropenem, and ciprofloxacin, yet retained susceptibility to polymyxin B. Moreover, except

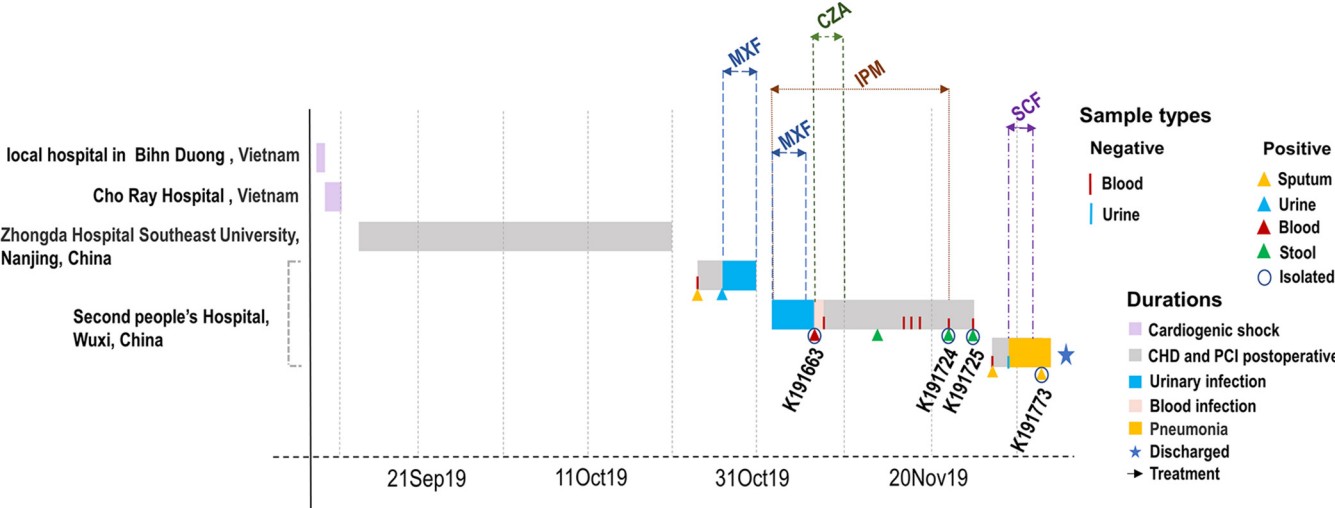

**FIG 1** Timeline of the patient's hospitalization, including diagnosis, the outcome of microbiological test, and antibiotic treatment. CHD, coronary heart disease; PCI, percutaneous coronary intervention; MXF, moxifloxacin; IPM, imipenem; CZA, ceftazidime-avibactam; SCF, cefoperazone-sulbactam.

for K191663, all isolates showed resistance against CZA and susceptibility to tetracycline. On the other hand, all clinical isolates were tigecycline non-susceptible *K. pneumoniae* in line with the EUCAST.

WGS data revealed that four strains belonged to *K. pneumoniae* ST16 which have been considered with high pathogenicity and high mortality (25). Phylogenetic tree based on single-nucleotide polymorphisms (SNPs) of four strains confirmed that K191724, K191725, and K191773 were all highly related (Fig. 2), all clustering in the same clade and closely related to the *K. pneumoniae* strain 18–2374 (accession no. NZ_CP041927.1)

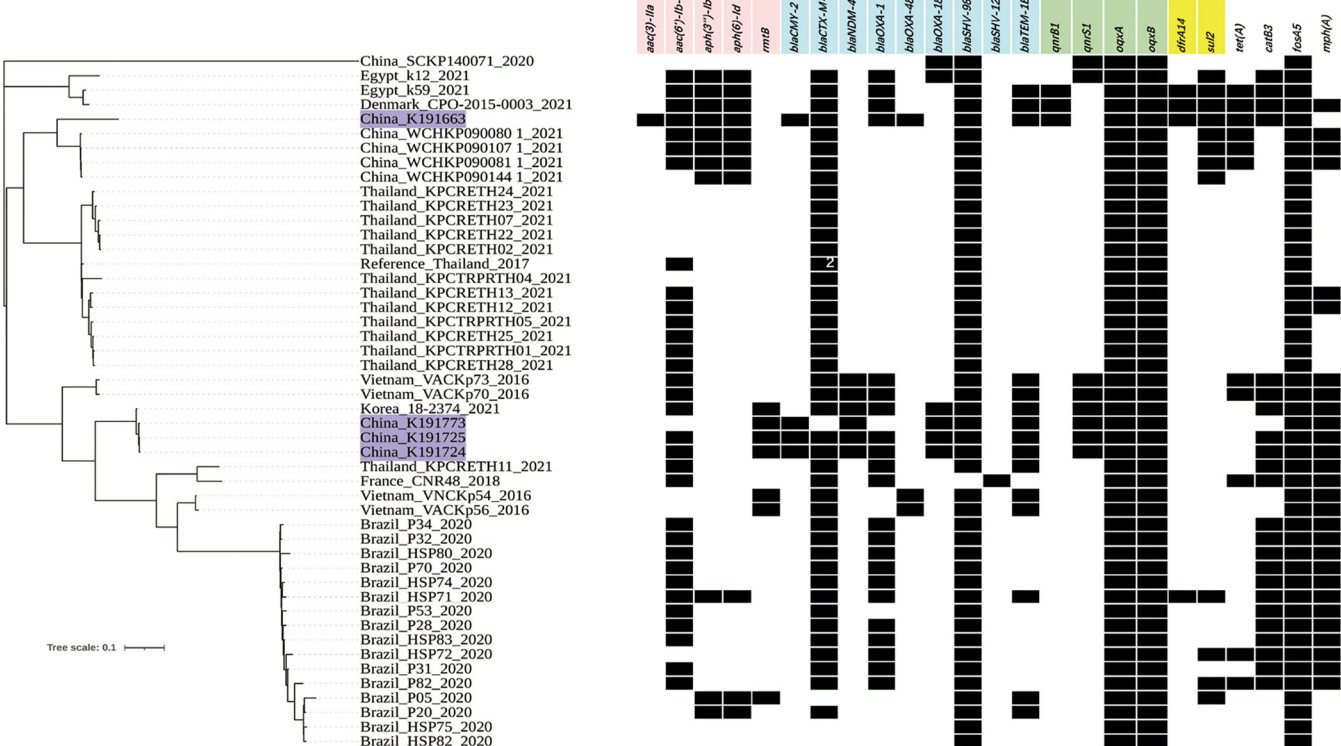

**FIG 2** Phylogenetic tree based on single-nucleotide polymorphism analysis of global 48 *K. pneumoniae* ST16. The presence of resistance genes contained in the involved strains was indicated with black squares. Antibiotic classes were shown above the resistance genes in different colors. Four strains included in this study were highlighted with a purple background. The integer in the square indicates the number of genes in the genome/plasmids.

isolated from sputum cultures of a chemotherapy patient in South Korea (7). However, K191663 clustered with another branch containing four NDM-5-encoding *K. pneumoniae* ST16 in China (15). Furthermore, comparative genomics suggested that four isolates carried different combinations of carbapenemases-encoding genes (K191663: $bla_{OXA-48}$; K191724, K191725, and K191773: $bla_{NDM-4}+bla_{OXA-181}$). In addition, analysis of WGS data showed that a two-amino-acid duplication (Asp137Thr138) in the L3 loop of OmpK36 was detected in all isolates, which was previously associated with ertapenem resistance and decreased susceptibility to meropenem and ceftazidime (26, 27).

Noted that the strain K191663 also differed from the CZA-resistant strains in virulence determinants (Table 2). All isolates carried virulence factors including type 3 fimbriae (*mrk*), type 1 fimbriae (*fim*), iron acquisition systems enterobactin (*entA/B/C/D/E/F/S*), salmochelin (*iroE/N*) and aerobactin (*iutA*), whereas a *Yersinia* high-pathogenicity island (HPI) which harbored *irp1/2*, *ybtA/E/P/Q/S/T/U/X*, and *fyuA* were only detected in K191663. According to the results of Kleborate (28), the yersiniabactin locus (*ybt* locus) and associated integrative conjugative element (ICE*Kp*) of this strain were *ybt9* and ICE*Kp3*, respectively. According to the combination of alleles, the yersiniabactin sequence type of K191663 was assigned to YbST 174. As a transcriptional regulator of AcrAB efflux pump, AcrR plays a pivotal role in drug resistance and virulence (29, 30). In our study, a frameshift mutation resulting from 1-bp deletion (G) in codon 139 was detected in K191663, whereas 4-amino acid deletion was found in the CZA-resistant strains (Thr144Leu145Lys146Glu147, refer to GenBank accession no. CP000647).

**Plasmid genome comparison of CZA-resistant isolates.** To analyze the dissimilarities between the CZA-resistant clinical strains isolated from distinct anatomical sites, the MICs and resistance genes of these cultures were further compared. Plasmid Multi-Locus Sequence Typing (MLST) analysis revealed similar incompatibility groups of plasmids carried by K191724, K191725, and K191773, and the MICs were almost identical between these isolates, except a 2-fold increase in MIC of chloramphenicol was found in K191773. Meanwhile, the absence of resistance genes such as *mphA*, $bla_{OXA-1}$, $bla_{CTX-M-15}$, *aac(6')-Ib-cr*, and *catB3* in K191773 raised our concern. Based on BLAST searches, we found pFII-191773 in the K191773 showed a high similarity with pSECR18-2374A, a plasmid harbored in the *K. pneumoniae* strain 18–2374 (coverage 95%, sequence identity 100%, accession no. NZ_CP041928.1), despite a 12-kbp fragment contained above-mentioned genes were detected in pSECR18-2374A. We designed primers to connect the contigs that contained those resistance genes in K191725. Sequence analysis highlighted that it was one of the plasmid fragments possessing a multidrug-resistant (MDR) region (MRR), downstream of a *repA* gene related to the IncFII family and consisted of four IS*26* elements flanking the following three segments: (i) IS*26-mph(A)-mrx-mphR(A)*-IS*6100*, (ii) IS*26-aac(6')-Ib-cr-bla$_{OXA-1}$*-*catB3*-IS*26*, and (iii) *bla$_{CTX-M-15}$-wbuC*-Tn*3*-IS*26*, revealed that the gene transfer of drug resistance genes mediated by IS*26* was present in our isolates (Fig. 3). Remarkably, no identical direct repeat (DR) was found around inverted repeats (IRs) of IS*26* on both sides of the fragment, suggesting that the homologous recombination occurred in the IS*26*-mediated translocatable unit.

**Characterization of plasmids carrying drug resistance genes in CZA-resistant strains.** We hypothesized that plasmids contribute to the differences in antimicrobial susceptibility of the *K. pneumoniae* isolates. In conjugative assays with azide-resistant *E. coli* J53 as the recipient, analysis of PCR products from transconjugants demonstrated that $bla_{NDM-4}$-carrying plasmid (pNDM4-191773, accession no. CP080366) in K191773 was self-conjugative. At the same time, the MICs of the conjugant J73-NDM4 were very similar to those of the donor bacteria. Under the threshold for minimum 90% identity and minimum 80% coverage, we detected the incompatibility group of pNDM4-191773 was IncFII(Yp) (90.84% identity). This 85-kb plasmid exhibited high similarity to the plasmid pSECR18-2374C from the *K. pneumoniae* strain 18–2374 (coverage 100%, sequence identity 99.99%, accession no. CP041930.1) (7) and pNDM1_045001 found in Sichuan, China (coverage 98%, sequence identity 100%, accession no. CP043383.1). The identical genetic organization of region surrounding the $bla_{NDM-4}$ (IS*91-groEL-groES-cutA-dsbD-trpF-ble*$_{MBL}$-$bla_{NDM-4}$-ΔIS*ba125*) was similar to the previously

**TABLE 2** Whole-genome sequencing analysis of isolates in this study

| Isolate | Sample | MLST | K type (wzc) | Replicon | Size (bp) | Incompatibility groups[b] (incs) | Carbapenemase encoding-genes[c] | Other resistance genes[c] and mutations | Virulence genes[d] | GenBank accession no. |
|---|---|---|---|---|---|---|---|---|---|---|
| K191663 | Blood | 16 | K51 | Chromosome | 5,361,205 | | | $bla_{SHV-98}$, $fosA5$ OmpK36V[e], AcrR[Δf] | $entA/B/C/D/E/F/S$, $iutA$, $iroE/N$, $acrA/B$, $vipA/tssB$, $fyuA$, $irp1/2$, $ybtA/E/P/Q/S/T/U/X$ | CP080353 |
| | | | | pOXA1-191663 | 136,344 | IncR-IncFIIK_7 | | $catB3$, $aac$ (6')-Ib-cr, $aac$ (3)-IIa, $dfrA14$, $aph$ (3'')-Ib, $qnrB1$, $aph(6)$-Id, $bla_{TEM-1B}$, $bla_{CTX-M-15}$, $tetA$, $sul2$ | | CP080359 |
| | | | | pFIB-191663 | 111,238 | IncFIB(pKPHS1) | | | | CP080358 |
| | | | | pOXA48-191663 | 63,589 | IncL | $bla_{OXA-48}$ | | | CP080360 |
| | | | | pX3-191663 | 38,157 | IncX3 | | | | CP080361 |
| K191724[a] | Stool | 16 | K51 | | 5,652,972 | IncFIB(K), IncFIB(pKPHS1), IncFIIK_5, ColKP3, Col440I, IncFII(Yp) | $bla_{NDM-4}$, $bla_{OXA-181}$ | $rmtB$, $aac(6')$-Ib-cr, $bla_{SHV-98}$, $bla_{TEM-1B}$, $bla_{CTX-M-15}$, $fosA5$, $mphA$, $qnrS1$, $catB3$ OmpK36V, AcrR* | $entA/B/C/D/E/F/S$, $iutA$, $iroE/N$, $acrA/B$ | JAHYSC000000000 |
| K191725[a] | Stool | 16 | K51 | | 5,801,062 | IncFIB(K), IncFIB(pKPHS1), IncFIIK_5, Incl1-I(Alpha), IncX3, ColKP3, Col440I, IncFII(Yp) | $bla_{NDM-4}$, $bla_{OXA-181}$ | $rmtB$, $aac$ (6')-Ib-cr, $bla_{SHV-98}$, $bla_{TEM-1B}$, $bla_{CMY-2}$, $bla_{CTX-M-15}$, $fosA5$, $mphA$, $qnrS1$, $catB3$ OmpK36V, AcrR* | $entA/B/C/D/E/F/S$, $iutA$, $iroE/N$, $acrA/B$ | JAHYSB000000000 |
| K191773 | Sputum | 16 | K51 | Chromosome | 5,306,347 | | | $bla_{SHV-98}$, $fosA5$ OmpK36V, AcrR[g] | $entA/B/C/D/E/FF/S$, $iutA$, $iroE/N$, $acrA/B$ | CP080362 |
| | | | | pFII-191773 | 190,150 | IncFIBK-IncFIIK_5 | | | | CP080363 |
| | | | | pFIB-191773 | 110,643 | IncFIB(pKPHS1) | | | | CP080364 |
| | | | | pCMY2-191773 | 95,539 | Incl1-I(Alpha) | | $bla_{CMY-2}$ | | CP080365 |
| | | | | pNDM4-191773 | 85,191 | IncFII(Yp) | $bla_{NDM-4}$ | $rmtB$, $bla_{TEM-1B}$ | | CP080366 |
| | | | | pOXA181-191773 | 51,479 | ColKP3-IncX3 | $bla_{OXA-181}$ | $qnrS1$ | | CP080367 |
| | | | | pCol440I-191773 | 4687 | Col440I | | | | CP080368 |

[a]Draft genome sequences.
[b]Using PlasmidFinder 2.1 and pMLST 2.0.
[c]Using ResFinder 4.1.
[d]Using Kleborate v.2.2.0.
[e]OmpK36V, OmpK36 variant with Asp137 and Thr138 insertions.
[f]Δframeshift mutations resulted from 1-bp deletion (G) in codon 139. (Refer to GenBank accession no. CP000647).
[g]4-amino acids deletion in AcrR (Thr144Leu145Lys146Glu147).

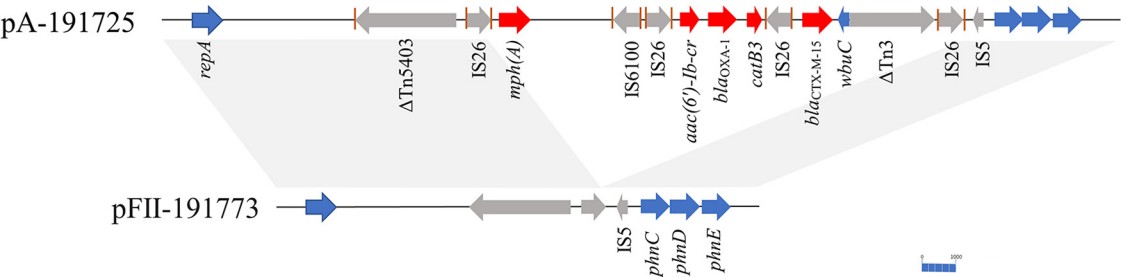

**FIG 3** Sequence alignment of pFII-191773 with the contig of pA-191725 carried *repA* gene related to IncFII family. The orange line segment represented the terminal inverted repeat (IR). Light-gray shading represents the homology region (100% nucleotide identity).

described (8), suggesting a conservative structure. The resistance genes $bla_{TEM-1B}$ and *rmtB* were located downstream of $bla_{NDM-4}$, flanked by IS26 and ΔTn3, and the latter comprised a Tn2 transposon variant with gene $bla_{TEM-1B}$. Furthermore, compared with the pNDM1_045001 carried by an *E. xiangfangensis*, in addition to the amino-acid substitution (Leu154Met) in the NDM-1, an insertion sequence, IS*Kox3*, was detected upstream of the *umuD* gene (Fig. 4).

Besides pNDM4-191773, a $bla_{OXA-181}$ and *qnrS1* co-existing plasmid (pOXA181-191773, accession no. CP080367) of 51,479 bp in size was also identified, harboring IncX3- ColKP3 fused replicon. Notably, transconjugants only possessing $bla_{OXA-181}$ were not obtained, whereas the co-transfer of $bla_{NDM-4}+bla_{OXA-181}$ and $bla_{NDM-4}+bla_{OXA-181}+bla_{CMY-2}$ were detected. Thus, we speculated that the transfer of pOXA181-191773 required the presence of other helper plasmids. As a widespread β-lactam resistance gene originated in India, similar genetic environments around $bla_{OXA-181}$ were also characterized in plasmids from other countries, such as Portugal (31) and Canada (32). The relatedness among pOXA181-191773 and other $bla_{OXA-181}$-harboring plasmids (*n* = 19) from the GenBank database were shown by a phylogenetic neighbor-joining tree (Fig. S1 in the supplemental material). IncX$_3$ was one of the narrow-host-range plasmids and mainly transmitted in *Enterobacteriaceae* (33). From our data, most of the similar plasmids come from *E. coli*, like p1-Ec-BERN-042 (CP042935.2), isolated from ST410 *E. coli* in Switzerland (34). This suggested that the same plasmid carrying $bla_{OXA-181}$ has spread in Europe.

**Structure of the extended-spectrum β-lactamase-encoding plasmid.** We characterized K191663 as a MDR bacterium in the absence of $bla_{NDM-4}$, its success may partly relate to 15 genes associated to the resistance to β-lactams [$bla_{SHV-98}$, $bla_{CTX-M-15}$, $bla_{OXA-1}$, $bla_{OXA-48}$, $bla_{TEM-1B}$], aminoglycosides [*aac(3)-IIa*, *aac(6′)-Ib-cr*, *aph(3″)-Ib*, *aph(6)-Id*], fluoroquinolones (*qnrB1*), sulfonamides (*sul2*), tetracyclines (*tetA*), chloramphenicol (*catB3*), fosfomycin (*fosA5*) and trimethoprim (*dfrA14*) (Table 2). In strain K191663, a self-conjugative plasmid pOXA1-191663 (accession no. CP080359) was detected for its broad range of antimicrobial resistance. However, unlike pNDM4-191773, pOXA1-191663 was susceptible to carbapenems and CZA. According to the WGS analysis, pOXA1-191663 was a multi-replicon (IncR-IncFIIK_7) plasmid of 136,344 bp in length with a GC content of 52.49% and was similar with the pKPC2_090515 (coverage 82%, sequence identity 99.86%, accession no. CP073288.1). At the same time, the MRR of pOXA1-191663 was ~44kb in size and identical in sequence to pG4584_136.4Kb from *K. quasipneumoniae* in the USA (accession no. CP034131.1) and pRHBSTW-00365_3 from *K. oxytoca* in the United Kingdom. In the recombination "hot spot" of pOXA1-191663, a Tn2 transposon variant was composed of $bla_{TEM-1B}$ and a transposase, disrupted by a conserved segment (IS*Ecp1*-$bla_{CTX-M-15}$-*wbuC*), followed by a cassette array with two flanking IS26 elements, including the $bla_{OXA-1}$, *aac(6′)-Ib-cr* and *catB3* genes. In addition, a 3-kb integron region containing an *intI1* and a gene cassette that carried the *dfrA14* was as previously described (35), and located upstream of *qnrB1*, bracketed by IS26 and ΔIS*3000*. Although some regions that carried resistance genes were in the opposite orientation in pOXA1-

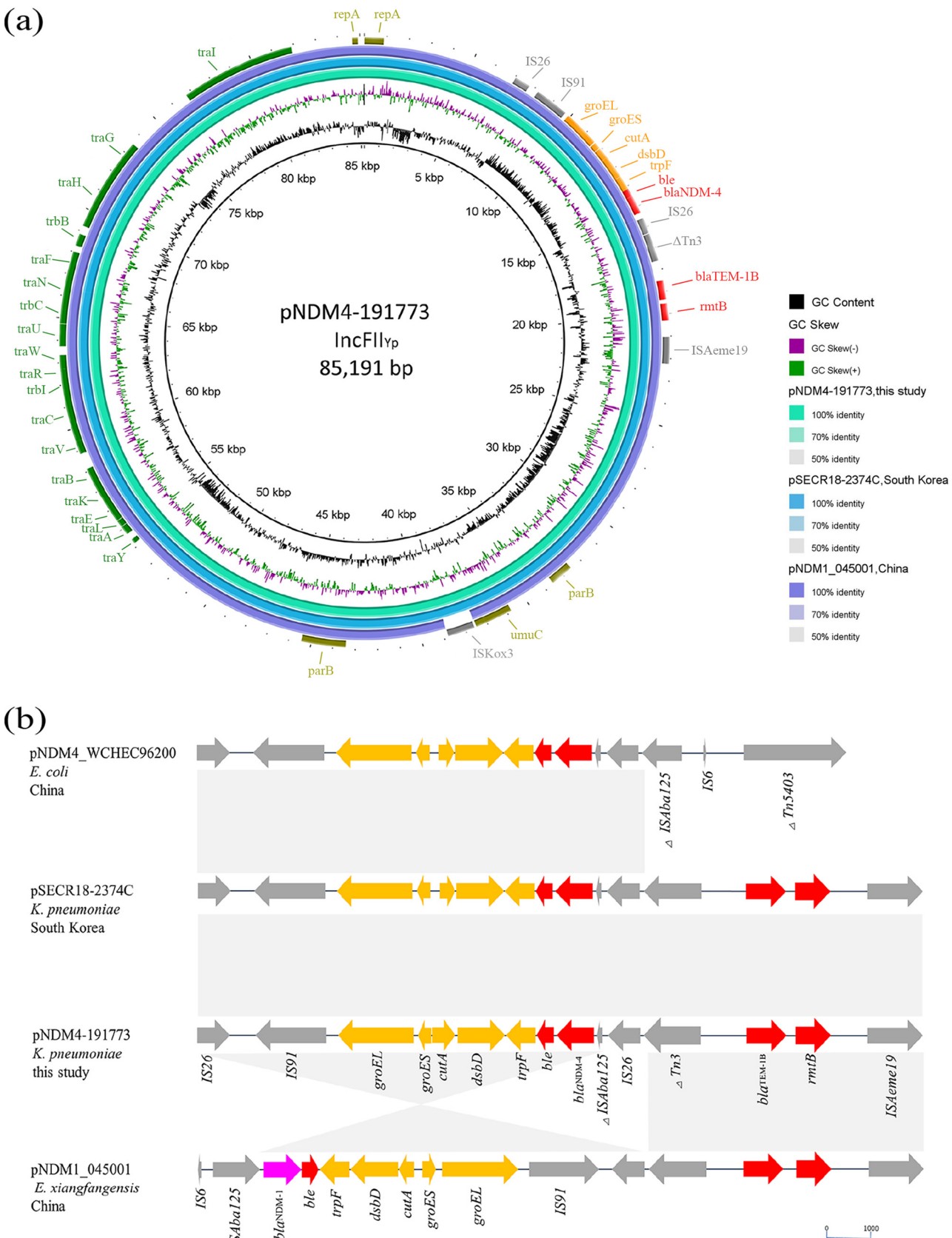

**FIG 4** Graphical representation of *bla*NDM-4-carrying plasmids sequence comparison. a, Circular map of the plasmid pNDM4-191773 and comparative genomics analysis with its similar plasmids. Starting from the center: (1) GC content of pNDM4-191773 with an average of 54.03% (used as reference).

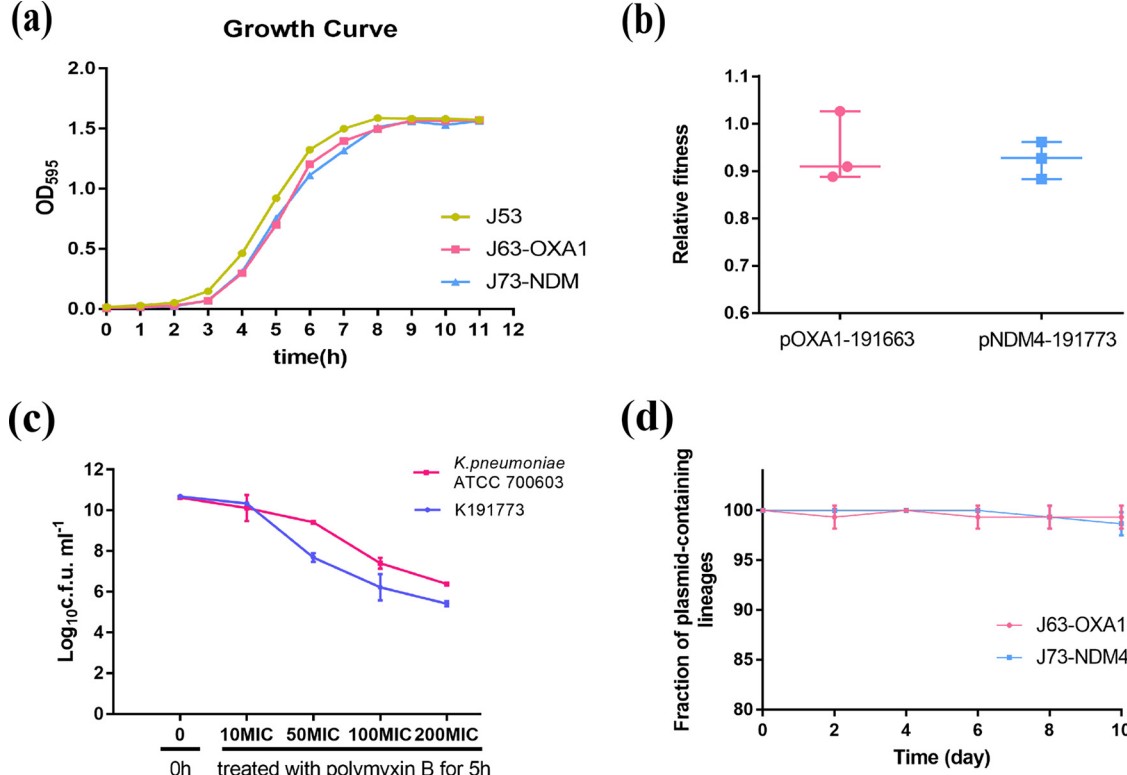

**FIG 5** Fitness costs and stability of pOXA1-191663 and pNDM4-191773 in *E. coli* J53 strain. a, The growth curve of the transconjugants and recipient J53. b, Competition experiments to assess the plasmids' relative fitness (w) upon *E. coli* J53. c, Killing kinetics of K191773 exposed to different concentrations of polymyxin B. d, Plasmid stability experiment results. All experiments were conducted in triplicate. Error bars denote the means (middle lines) and standard deviations.

191663, high similarity indicated that each was discernible as a distinct DNA mobility event (Fig. S2 in the supplemental material).

**Fitness costs and stability of two conjugative plasmids.** Since *K. pneumoniae* ST16 has been related to nosocomial outbreaks, in order to investigate the effect of temperature on plasmids transferability, we performed the conjugation tests at 25°C and 37°C, respectively. Notably, the transfer frequencies of pOXA1-191663 and pNDM4-191773 were $(9.238 \pm 0.502) \times 10^{-5}$ and $(5.835 \pm 0.693) \times 10^{-1}$ per recipient at 25°C, compared to $(4.96 \pm 0.594) \times 10^{-5}$ and $(2.42 \pm 0.334) \times 10^{-2}$ per recipient at 37°C. Our data showed that both plasmids have relatively higher conjugation frequencies at room temperature ($P < 0.05$).

After 10-days serial passage without antibiotic treatment, both J63-OXA1 and J73-NDM4 showed an insignificant plasmid loss (98.67%-99.33% fraction of plasmid-containing lineages) (Fig. 5d). Plasmid stability is concerned with any effects the plasmid might have on bacterial fitness (36). Consequently, we evaluated the effect of acquiring resistant plasmids on biological fitness and observed no obvious differences in the growth rate among the recipient strain J53 and the transconjugants (Fig. 5a). In addition, compared with neutrality (w = 1.0), the results of competition experiments suggested different biological cost between pNDM4-191773(w = $0.9238 \pm 0.02256$, $P = 0.03$) and pOXA1-191663(w = $0.9412 \pm 0.04297$, $P > 0.05$) carriage (Fig. 5b).

**FIG 4** Legend (Continued)
(2) GC skew (G-C/G+C), with a positive GC skew toward the inside and a negative GC skew toward the outside. (3) pNDM4-191773 plasmid sequence (CP080366). (4) pSECR18-2374C (CP041930.1). (5) pNDM1_045001 (CP043383.1). (6) Gene annotation. Red, antibiotic resistance; green, transfer conjugation; gray, insertion sequence; olive, replicon and backbone gene; orange, *bla*~NDM-4~ surrounding gene. b, A detailed linear comparison of the genetic organization of region surrounding the *bla*~NDM-4~, *bla*~TEM-1B~, and *rmtB*. Arrowhead indicates the direction of transcription. Light-gray shading represents a similar region (>99% nucleotide identity). A scale of 1000 bp is attached to the corner.

**The persister formation of K191773.** Since K191773 was isolated from the patient after re-hospitalization following a recurrent infection, persister assays were performed to investigate the persister formation of K191773. Compared with the higher MICs of carbapenems (both MIC of imipenem and meropenem $>$ 128 mg/L), polymyxin B with lower MIC (0.5 mg/L) was selected for killing kinetics under different concentrations. The results showed that K191773 survived with $\sim 10^5$ CFU mL$^{-1}$, while *K. pneumoniae* ATCC 700603 could stay with $\sim 10^6$ CFU mL$^{-1}$ at 200-fold MIC of polymyxin B for 5 h (Fig. 5c).

## DISCUSSION

According to a meta-analysis of therapeutic efficacy following antibiotic treatment, the pooled mortality of patients with CRKP infections was 37.2% (37). As one of the critical priority pathogens classified by the World Health Organization (WHO), further research and development of new antibiotics for this kind of hospital-acquired strain are urgent (38). In this study, four carbapenem-resistant cultures obtained longitudinally at several body sites belonged to *K. pneumoniae* ST16, which is not part of the major clonal complex in China. These strains can be classified as CZA-susceptible and CZA-resistant bacteria based on the AR profiles. Phylogenic analysis suggested that more than one subpopulation of *K. pneumoniae* ST16 was present in the same patient. Given the modest rate of within-host point mutation for *K. pneumoniae* ($\sim$10 SNPs/ year) (39), we hypothesized that it is a selective advantage under antibiotic therapy that leads to the emergence of different subpopulations in sequential isolates. Although the CZA-susceptible strain (K191663) was obtained from blood, it was in a cluster with four samples of *K. pneumoniae* associated with nosocomial flora in China (15), suggesting a common ancestor. Previously, *ybt* operon, a yersiniabactin-encoding system related to immune evasion, had a high carriage rate (100%, 15/15) in carbapenemase-producing *K. pneumoniae* ST16 isolated from rectal swab samples in Thailand (40). However, *ybt9* (located on ICE*Kp3*) was only detected in K191663 in this study, indicating that collecting different specimen types for analysis during infection contributes to discovering the diversity of bacterial genomes.

Nosocomial outbreaks caused by *K. pneumoniae* ST16 carrying *bla*$_{NDM-4}$ or *bla*$_{OXA-181}$ have been reported in Vietnam, where our patient returned. However, they did not have a significant correlation with our CZA-resistant isolates. Unexpectedly, a *K. pneumoniae* strain(18-2374) reported in South Korea was closely related to the two CZA-resistant strains isolated from feces (7). The most notable difference between these strains was the *bla*$_{CMY-2}$-carrying plasmid detected in K191724 and K191725. Our patient had no travel history to South Korea, and it is unknown how they became infected with a nearly identical clone. We do not exclude the possibility that this NDM-4- and OXA-181-producing *K. pneumoniae* ST16 has been widespread, although the relevant reports were rare. The previous reports had shown the disruption of an outer membrane porin mediated by IS*26* with a tandem array of $\beta$-lactamase genes, which result in non-carbapenemase-producing carbapenemase-resistant *Enterobacteriaceae* emergence (41). In contrast, this study identified the mutations of outer-membrane porin for the amino-acid duplication in OmpK36. Noteworthily, the loss of 12-kbp fragment in K191773 further implied the mobilization of resistance genes by IS*26*. The patient's imipenem therapy was discontinued after detecting *bla*$_{NDM-4}$, *bla*$_{OXA-181}$, and *bla*$_{OXA-1}$-coexisting *K. pneumoniae* from feces (K191724). The strain K191773 was eventually isolated and lacked the MRR flanked by two copies of IS*26* compared with K191725. Both copies of the IS*26* were in the same orientation. However, one of the flanking IS*26* stayed in the pII-191773, implying that a translocatable unit containing the resistance genes and one IS*26* appears to be excised from the plasmid without the pressure of antibiotic screening. It has been suggested that compensatory evolution involving the large-scale plasmid deletions may counteract the fitness costs and improve the plasmid stability (18). Perhaps acquiring a broader-spectrum resistance gene, like *bla*$_{NDM-4}$, weeded out relatively narrow-spectrum resistance genes to increase fitness (42). Interestingly, our results showed that K191773 had a

lower level of persisters tolerant to a high concentration of polymyxins B than *K. pneumoniae* ATCC 700603. However, continuous monitoring is necessary to determine whether the lost fragment of plasmids is recombined again and whether its occurrence is related to specific anatomical sites.

Through the analysis of conjugants, we found that the $bla_{NDM-4}$-containing plasmid (pNDM4-191773) presented multi-drug resistance. Similar to the $bla_{NDM-4}$-harboring IncX3 and IncFIA plasmid carried by *E. coli* which have been reported in China previously (8, 9), pNDM4-191773, an IncFII(Yp)-like plasmid that produced NDM-4, showed the conservative flanking sequence and mobile elements around $bla_{NDM-4}$ (IS*91-groEL-groES-cutA-dsbD-trpF-ble*$_{MBL}$-$bla_{NDM-4}$-ΔIS*ba125*). IncFII(Y)-type plasmids were related to the worldwide spread of $bla_{NDM-1}$ within *K. pneumoniae* in South Africa (43), *Leclercia adecarboxylata* in Mexico (44), and $bla_{VIM}$-carrying *K. grimontii* in Switzerland (45). To note, the transfer frequency of pNDM4-191773 was higher at room temperature, indicating that it carries a transmission risk in the hospital environment. Despite the lower conjugation frequency, pOXA1-191663 had a similar characteristic. In contrast, pNDM4-191773 had a statistically significant fitness reduction *in vitro* and was more unstable than pOXA1-191663. These available data suggested that pNDM4-191773 can spread easily to other hosts by conjugation, albeit at some biological costs. In addition, multiple insertion sequence elements were found scattered throughout pOXA1-191663, which may facilitate the dissemination of the antimicrobial resistance determinants. Last year, we reported KPC-2 and NDM-5 co-producing CRKP ST11 isolated from a clinical patient in Wuxi, China (11), and subsequently, Xi Li et al. identified the MBL double-positive ($bla_{VIM-4}$ and $bla_{NDM-1}$) plasmid in *Klebsiella michiganensis* (46). These researchers of two carbapenem-resistant genes carried by the same strain increase the awareness of carbapenem-resistant *Enterobacteriaceae*. To the best of our knowledge, this is the first report on the coexistence of $bla_{NDM-4}$, $bla_{OXA-181}$ in *K. pneumoniae* ST16 in China, to further highlight the importance of surveilling the prevalence of NDM enzymes and their latent vectors.

In summary, we analyzed the within-host genomic dynamics of four *K. pneumoniae* ST16 isolates sequentially isolated from a single patient with long-term exposure to antibiotic therapy. The $bla_{NDM-4}$-harboring plasmid is of concern for the efficient conjugation capacity at both 25°C and 37°C. Further studies on resistance gene transfer and plasmid evolution mediated by insertion sequences are crucial for clinical infection control.

## MATERIALS AND METHODS

**Case report and bacterial strains.** On 24 October 2019, a 45-year-old patient presented to the Affiliated Wuxi No.2 People's Hospital in Wuxi, China, for coronary heart disease and anterior wall acute myocardial infarction. He had been hospitalized in the intensive care unit of Zhongda Hospital, Southeast University in Nanjing, China, after returning from Vietnam with the diagnosis of cardiogenic shock and deep coma. Then, he was treated with a tracheotomy and anti-infective therapy. The patient was admitted to the Affiliated Wuxi No.2 people's hospital three times. During the first hospitalization, sputum and urinary cultures grew carbapenems-susceptible *K. pneumoniae*, his symptoms relieved with moxifloxacin treatment. However, he had a recurrence of urinary tract infection after discharge and carried out a re-admission on 2 November. Imipenem was added as a combined treatment with moxifloxacin for infections. On 7 November, blood cultures grew a multidrug-resistant (MDR) *K. pneumoniae* (K191663). Subsequently, the patient was treated with ceftazidime-avibactam for 3 days in addition to imipenem. Imipenem was administered until the 22 November, which culture of a fecal sample on this day grew a CRKP strain (K191724), and fecal culture was positive again on 25 November (K191725). Two days later, the patient was readmitted with bacterial pneumonia and treated empirically with cefoperazone-sulbactam for 3 days. Later, CRKP (K191773) was isolated in sputum culture on 3 December (Fig. 1). After symptomatic treatment, the patient was finally cured and discharged.

**Antimicrobial susceptibility testing.** The broth microdilution methods were used to investigate the antibiotic susceptibility of the strains *in vitro* according to Clinical and Laboratory Standards Institute guidelines (47). *Escherichia coli* ATCC 25922 and *K. pneumoniae* ATCC 700603 were used as the quality control strains, and the interpretation of the results was based on 2020 CLSI M100 30th Edition breakpoints except for the tigecycline, which referred to the EUCAST v.11.0 breakpoints (https://eucast.org/).

**WGS and analysis.** All isolates were sequenced using an Illumina HiSeq to generate 150 bp paired-end reads and 100× coverage, while some isolates (K191663 and K191773) were analyzed using PacBio RSII analysis to obtain complete chromosome and plasmid sequences (Majorbio Co., Ltd. Shanghai, China). We used SOAPdenovo v.2.04 (48) and Unicycler v0.4.6 (49) for *de novo* assembly in the draft and

complete genome sequence. Gene annotations were carried out with the RAST online tool (http://rast.theseed.org/FIG/rast.cgi) and corrected via BLASTn (https://blast.ncbi.nlm.nih.gov). MLST was analyzed by MLST v.2.0 (https://cge.cbs.dtu.dk/services/MLST/), while the capsular type of *Klebsiella* strains was determined according to *wzc* gene sequence (50). Plasmid replicon types, plasmid MLST, and resistome were also identified by CGE server (https://cge.cbs.dtu.dk/services/, with the threshold for minimum 90% identity and minimum 80% coverage). Resistance genes *aac(6′)-Ib-cr*, *bla*$_{NDM-4}$, *bla*$_{CMY-2}$, *bla*$_{OXA-1}$, and *catB3* of strain K191724 (omission in draft genome sequence), mutations in *ompK36* and *acrR* from all strains were verified by PCR amplicon sequencing with the primers listed in Table S1 in the supplemental material. The ISfinder Database (https://www-is.biotoul.fr/, default parameters) was used to name the insertion sequences. The BLAST Ring Image Generator (BRIG) tool served to generate the circular map of plasmids and compared the plasmids against the highly homologous plasmids in the NCBI database.

**Phylogenetic analysis.** Single-nucleotide polymorphism (SNP) analysis was performed by the kSNP 3.0 based on concatenated genome sequence data of 48 *K. pneumoniae* ST16 isolates, including 9 Chinese ST16, 39 international ST16 (51). A phylogenetic tree of the genome and a phylogenetic neighbor-joining tree of *bla*$_{OXA-181}$-carrying plasmid were constructed by MEGA X v.10.2.5 (52) and later visualized using the iTOL website (https://itol.embl.de/).

**Conjugation assay and plasmid stability test.** Conjugation assays were carried out using azide-resistant *E. coli* J53 as a recipient at both 25°C and 37°C in Luria-Bertani (LB) broth (53). The transconjugants were selected on LB agar plates containing sodium azide (100 mg/L) with meropenem (0.3 mg/L) or tetracycline (5 mg/L), respectively. The resistance genes *bla*$_{NDM-4}$, *bla*$_{OXA-181}$, *bla*$_{CMY-2}$, *tet(A)*, and 16sRNA of conjugants were amplified and sequenced by PCR with the primers in Supplement Table S1 in the supplemental material.

Plasmid-containing conjugants were subjected to plasmid stability test as previously described (36). Briefly, three independent lineages of each plasmid-containing strain were cultured overnight at 37°C in 1 mL of LB broth supplemented with meropenem (0.3 mg/L) or tetracycline (5 mg/L). These samples were washed and resuspended in antibiotic-free LB broth. Serial passaging of 1 $\mu$L of the overnight culture to 1 mL LB broth was performed daily. For every 2 days, cultures were diluted and plated on antibiotic-free LB agar (LBA) plates. Then, 50 colonies from each lineage were randomly selected for *bla*$_{NDM-4}$ or *aac(6′)-Ib-cr* target PCR to verify the plasmids stability.

**Growth curves assay and *in vitro* competition assay.** Growth curves assay was used to calculate the fitness of plasmid between the different transconjugants and *E. coli* J53 (54). Briefly, the overnight cultures were diluted at 1:50 in LB without antibiotics and incubated at 37°C in a shaking bath (200 rpm). Absorbance (OD$_{595nm}$) was measured every hour for 11 h, and each sample was repeated three times.

To compare each transconjugant's relative fitness (w) with that of the progenitor strain, we performed the *in vitro* competitive assays as previously described by Ke Ma et al. (55). Overnight cultures were adjusted to a 0.5 McFarland standard. An aliquot of 10 $\mu$L of competitors was mixed in 1 mL LB broth at a volumetric ratio of 1:1, which constituted the initial mixed culture (0 h), followed by culturing at 37°C for 24 h with shaking (200 rpm). The number of CFU growing on LBA containing 100 mg/L sodium azide with meropenem (0.3 mg/L) or tetracycline (5 mg/L) was subtracted from the number of CFU growing on LBA containing sodium azide to determine the number of plasmid-free cells in the mixed population. Calculating the relative fitness (w) using this formula, w = $\log_{10}(Ng_{24}/Ng_0)/\log_{10}(Nw_{24}/Nw_0)$, where *Ng* and *Nw* are defined as the number of plasmid-carrying and plasmid-free cells, respectively. A w <1 value suggests a fitness cost, while w >1 suggests a fitness advantage. Each experiment was independently repeated in triplicate.

**Persister assay.** To investigate the persister formation of the final isolate, K191773, under different concentrations of antibiotics. Stationary-phase bacteria were exposed to 10-, 50-, 100- or 200-times the MIC of polymyxin B (56). In addition, *K. pneumoniae* ATCC 700603 was chosen to be a control. Samples were incubated at 37°C for 5 h with shaking (200 rpm).

**Statistical analysis.** We used Student's *t* test with GraphPad Prism software 7.0 to compare the differences in transfer frequency and relative fitness. *P*-values < 0.05 were considered significant.

**Ethics.** This study has obtained ethical approval and informed consent of the patient. The ethical code is 20180309059.

**Data availability.** The complete sequences of the two strains in this study, K191663 and K191773, were deposited in the GenBank database (accession numbers. CP080353-CP080361 and CP080362-CP080368). The draft genome sequences of K191724 and K191725 were submitted to GenBank under accession numbers JAHYSC000000000 and JAHYSB000000000, respectively. Raw sequence reads are available on NCBI under Bioproject accession number PRJNA749815.

## SUPPLEMENTAL MATERIAL

Supplemental material is available online only.

**SUPPLEMENTAL FILE 1**, PDF file, 0.2 MB.

## ACKNOWLEDGMENTS

We thank Fei Liu from the CAS Key Laboratory of Pathogenic Microbiology and Immunology, Chinese Academy of Sciences, for constructing a phylogenetic tree based on SNPs analysis. Thanks to the Pathogenic biology platform of the Public Experimental Technology Center at Southwest Medical University.

This research was funded by the National Natural Science Foundation of China (31500114), the Sichuan Province Science and Technology project (2020YJ0338), and the Southwest Medical University Foundation (21YYJC0529). The funder had no role in study design, data collection, analysis, decision to publish, and preparation of the manuscript.

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
