## [Reviewer comments · Microbiology Spectrum]

Microbiology Spectrum

Comparison of two distinct subpopulations of *Klebsiella pneumoniae* ST16 co-occur in a single patient

Yingshun Zhou, Biying Zhang, Renjing Hu, Qinghua Liang, Shuang Liang, Qin Li, Jiawei Bai, Manling Ding, and Feiyang Zhang

Corresponding Author(s): Yingshun Zhou, Southwest Medical University

Review Timeline:

Submission Date:	December 17, 2021
Editorial Decision:	January 11, 2022
Revision Received:	March 11, 2022
Editorial Decision:	March 26, 2022
Revision Received:	March 31, 2022
Accepted:	March 31, 2022

Editor: Karen Carroll

Reviewer(s): Disclosure of reviewer identity is with reference to reviewer comments included in decision letter(s). The following individuals involved in review of your submission have agreed to reveal their identity: Clement K.M. Tsui (Reviewer #2)

Transaction Report:

DOI: <https://doi.org/10.1128/spectrum.02624-21>

January 11, 2022

Dr. Yingshun Zhou
Southwest Medical University
Zhongshan
Luzhou
China

Re: Spectrum02624-21 (Repeated isolation of NDM-4 and OXA-181-Producing ST16 *Klebsiella pneumoniae* from a single patient in China)

Dear Dr. Yingshun Zhou:

Thank you for submitting your manuscript to Microbiology Spectrum. Your paper has been reviewed by 5 experts in the field. The overall assessment is that the work is of interest to Spectrum readership. However, several of the reviewers recommend major revisions, including some additional genomic analysis. At least two of the reviewers would like significant reorganization of the paper. After you have reviewed the comments, please let me know if you wish to move forward with submission of a revised manuscript. Your revised manuscript will require another round of reviews.

Link Not Available

Sincerely,

Karen Carroll

Journals Department
Reviewer comments:

Reviewer #1 (Comments for the Author):

Biying Zhang et al repeated isolation and identified four strains of ST16 *Klebsiella pneumoniae* isolates from the same hospitalized patient and succeeded in assessing the differences between strains by whole genome analysis and phenotypic studies. The acquisition of blaNDM-4-harbored or blaOXA-181-harbored plasmid and IS26-mediated HGT indicated the changes in the bacteria as they adapt to antibiotic treatment. The GZA-susceptible strain had a stronger ability in biofilm formation. The clinical background was sufficiently well described and multiple methods were used to elucidate that the phenotype and antibiotic resistance genes of the same clone were significantly diverse in different periods of antibiotic treatment. The loss of some resistance genes revealed the bacterial adaptation strategies, which is beneficial to other groups studying the infectious prognosis and the source of persistent bacterial, not just those focused on the mechanism of bacterial resistance. The discussion linking the different analysis together was well explained. This study provides a reference for the in vivo evolution of

clinical strains. Overall, the paper is interesting, and data is well presented and well written.

Minor revision:

1. The font size of the figure of the clinical background can be larger.
2. Please pay attention to the spelling of "conjugant" in Table 1.
3. Line 220: "Plasmid MLST (pMLST) analyses identified the same Incs of plasmids in K191724, K191725 and K191773..." According to Table 2, the Incs in K191724 was different from K191725 and K191773. Please revise this part.
4. Line 228: "IS26- aac(6')-Ib-cr-blaOXA-1- catB3-IS26", the gene names here are inconsistent with the annotation in Figure 3.

Reviewer #2 (Comments for the Author):

Overall, this study provided interesting data on the repeated isolation of carbapenemases bearing *K. pneumoniae* ST16 clinical samples in a patient in China. The authors performed antimicrobial susceptibility testing (AST), conjugation/biofilm assays and whole-genome sequencing (WGS). They also compared their plasmid sequences to other related samples in other countries. However, the manuscript was not well written with quite a few errors and several statements/data lacked strong evidence or support.

Major questions:

1. In order to determine the origin of KP ST16 in China, could you do a genome comparison/phylogeny (chromosomal DNA) to the sample from Korea (line 184) and other parts of China (any additional ST16 in other cities)? Can you run Kleborate on your genome sequences to determine the potential virulence factors?
2. It was interesting to observe the loss of 12-kbp fragment in K191773 after stop using imipenem. What happen when K191773 is grown on media with different conc. of imipenem again?
3. Did the authors isolate other carbapenemase producing Enterobacterales from the same stool samples?
4. Fitness cost cannot be estimated based on growth kinetics (Fig. 5) alone. It would be better to perform competition assay between the parental and conjugated strains! What is the plasmid stability?
5. In the biofilm formation experiment, it is important to have controls! Otherwise, how to justify their abilities in colonization?

Minor corrects/clarification:

Any written consent from the patient/ IRB for this study?

The current title does not reflect the content/aims of the investigation, which appeared to illustrate the acquisition of plasmid in the maintenance of carbapenemase in KP ST16.

The abstract and "importance" will need some revisions to tighten the content. For example, line 26-27, this sentence was poorly written.

L69, add *K. pneumoniae* after clinical

L70, "deeply" means important/ significant?

L72, replace "reasons" with mechanisms; replace "reference" by "baseline information"

L87, What kind of "symptomatic treatment"?

L96, Why not a *Klebsiella pneumoniae* was used as a reference? *K. quasipneumoniae* is a different species.

L97, Which CLSI session?

L 101, remove "additionally"

L115, add reference for MEGA-X

L120, replace "amplificated" with "amplified"

L136, which software was used to run ANOVA?

L139-142, please add SRA #, BioProject #

Table 1. J53, J63, J73 should be listed at the bottom of the table.

Fig.2. Diagram 2a was generated using BRIG. Which software was used to generate diagram 2b?

Fig S3. Better to plot the tree as a phylogram and put the bootstrap values on the branch. Difficult to interpret the relationship of these samples.

L280, change "multi-resistance" to multi-drug resistance

L283, there was no strong evidence from the investigation to support "increase persistence over long periods"

L293, please rephrase "...can last longer under pressure .."

L294, rephrase " .. we reported, for the first time the coexistence.." as "this is the first report on the coexistence .."

L309, "free fitness cost" was bad description

L310, What kind of virulence factors?

L317, this is not a study on the evolution of ST16 in China.

Reviewer #3 (Comments for the Author):

The authors perform characterization and whole-genome sequencing of 4 *Klebsiella pneumoniae* isolates obtained from a hospitalized patient in China. The isolates undergo Illumina and PacBio sequencing, antibiotic susceptibility minimum inhibitory concentration testing, and biofilm characterization. The genomic assembly, annotation, and characterization identifies molecular determinants for carbapenem resistance (*bla*NDM, and *bla*OXA-181 mediated). Further, the use of conjugation assays with *E. coli* J53 demonstrate that carbapenem resistance is transferable. The plasmid and resistance gene analysis is impressive. However, the genomic characterization described is insufficient to classify the relatedness of the isolates. The authors use of ANI for this purpose is not suitable compared to the alternative (whole-genome alignment or core-genome alignment). Accordingly, it is not exactly clear whether this is persistence of the same clone or co-existence of two clones within the same sequence type. The biofilm characterization does not fit within the scope of this manuscript and should be removed for consideration.

Minor Comments

Line 65 - Define CPS (and all acronyms at first use)

Line 81 - Susceptible to what? Be specific

Line 120 - Do you mean primers instead of "premiers"?

Line 127 - Were the plates TC treated or not?

Lines 139-142 - I am able to see that the strains are deposited in NCBI. I would also recommend that the authors reference their assigned BioProject ID so that people interested in their strains have a single webpage to see what they sequenced.

Major Comments

Lines 151-152 & Line 181 -182 - In this context, ANI really should just be used to verify that the isolates are in fact *Klebsiella pneumoniae* (and not another member of the species complex). I would not support this statement without additional SNP analysis to verify clonality. I would recommend that the authors remove this from the text or perform more SNP analysis.

Lines 307-308 - This is a major overstatement. The study did not examine therapy outcomes in single versus combination therapy. This study examined *Klebsiella* isolates.

Lines 310-311 - This is too speculative, even for a discussion. Biofilm formation is a multifactorial process and it is difficult to attribute individual gene presence/absence to this phenotype.

Lines 312-313 - Please cite literature indicating that the *AcrAB* gene components are important for biofilm formation.

Reviewer #4 (Comments for the Author):

Zhang et al., performed an interesting study about the identification and molecular characterization of CPE from the same patient. This study is very well written and clear. It is very interesting to identify the same *Klebsiella pneumoniae* clone with different genetic and virulence backgrounds. The authors showed that *K. pneumoniae* ST16 possessed a series adaptation in response to antibiotic pressure. Even so, there are some points the authors should consider before the acceptance of the manuscript.

-Abstract: It could be interesting to indicate the detected mutations in *AcrR*.

-Introduction: L56, the NDM-4 was identified from Egypt, not from Japan, please correct.

-L80-L81: how did you know *K. pneumoniae* was susceptible when recovered from sputum and urinary cultures during the first hospitalization of the patient? And on what basis the patient had received moxifloxacin for the treatment?

-L80-L87: If possible, include the dates of the patient admission and bacterial isolation on the text to prevent any conflict.

-On Antimicrobial susceptibility testing, you used EUCAST v.11.0 breakpoints for tigecycline, is the methodology of CLSI guidelines are the same as EUCAST? If not, it is better to repeat the experiment for tigecycline totally based on EUCAST guidelines?

-Please include a reference to the conjugation assay.

-L152-L54, *bla*OXA-1 is not carbapenemases-encoding genes, please exclude and correct this sentence. Also correct it in Table 2.

-Table 1, please indicate the meaning of bold and underlined values.

- Also, in table 1 please indicate (R), (I), or (S), after each value to indicate resistant, intermediate, or sensitive, respectively, and to prevent confusion.

- Please add the MIC results of quality control strains in Table 1.

-L257-259 is very confusing, please explain again why you can not get a transconjugant with *bla*OXA-181

-L258, please correct *bla*OXY-181.

- In the discussion, please discuss in detail if NDM-4 and/or OXA-181 producing *K. pneumoniae* ST16 have been previously identified or not? Which country? The possibility of their dissemination between countries?

-L293-L294, please explain your meaning with supporting reference(s).

-Several CRE isolates co-producing different carbapenemases and co-infection of with multiple CRE in the same patient has been identified, please discuss.

-Finally, what about the ethical approval of this study?

Reviewer #6 (Comments for the Author):

Repeated isolation of NDM-4 and OXA-181-Producing ST16 *Klebsiella pneumoniae* strain from a single patient in China

In this manuscript, the authors analyze antibiotic susceptibility, antibiotic resistance genes, virulence genes, and mobile genetic elements in four *K. pneumoniae* clinical isolates. These isolates were taken longitudinally from a 45-year individual with multiple comorbidities which included urinary tract infection, culture positive blood stream infection, and culture positive pneumonia. The patient's treatment included multiple antibiotic courses with differing susceptibility results from the patient bacterial isolates.

The authors should be commended for presenting an interesting genomics study grounded in a clinically interesting case. Beta-lactams are some of the most used antibiotics in clinical settings and *K. pneumoniae* is a common hospital pathogen. The spread of carbapenem resistance should concern all clinicians worldwide. The authors should also be commended for laying out how some of their findings fit with findings from other groups. Unfortunately, in its present form, the sections of the manuscript do not fit together to form a cohesive whole.

1. The overarching claims of the manuscript are not succinctly laid out beyond that antibiotic resistant *K. pneumoniae* isolates were collected from a patient in China.
2. The Introduction does not provide adequate background for the claims made in the paper or set the reader up to understand where the authors are trying to lead with the subsequent Results and Discussion sections. Some sections of the Introduction are extraneous and critical pieces of information for understanding the background for the manuscript are missing. Other sections of the Introduction lead the reader to expect results or findings that are not addressed in the manuscript.
3. The Results section does not have strong descriptions of why the authors conducted the experiments, or how said experiments fit together to prove the overarching claims of the manuscript.
4. The Discussion does not flow from section to section, and it does not put the results into cohesive narrative. It also does not discuss the limitations of the study.

Though the manuscript contents are of interest to the readership of *Microbiology Spectrum*, this work requires major revisions.

Major Comments

The introductory paragraph needs to be rewritten to focus more on providing background for the scientific questions answered by the manuscript. For example, is the information on amino acid substitution for NDM-1 to NDM-4 relevant to the claims of this manuscript? In the introduction, the authors state, "Understanding the mechanism of resistance gene acquisition and stability could enhance our understanding of the evolution of clinical strains." However, they do not address the mechanism of gene acquisition or stability in the manuscript.

After finishing the introduction, the reader should be able to anticipate what results will be shown in the following sections. For example, in the Biofilm section of the results, the authors say, "Biofilm experiment of isolates were carried out to investigate their ability to colonization." However, biofilms and colonization are not mentioned in the introduction and no references are given. Try to tie in the different results sections to the introduction and provide the basic rationale for conducting the analysis presented in the manuscript.

In the discussion, the authors say that they report "for the first time the co-existence of blaNDM-4 and blaOXA-181 in ST16 *K. pneumoniae* in China, to highlight the importance of surveillance [sic] the prevalence of NDM enzymes and its latent vectors." However, *K. pneumoniae* sequence types and their global distribution are not touched on in the Introduction. The authors should introduce sequence types and why the reader should care about them in the Introduction. Specifically, they should provide any background for the reader to understand why they should care that ST16 is now in China and where ST16 is currently prevalent. Can the authors comment on the prevalence of Clone ST16 in Vietnam? The paper seems to suggest that the individual acquired the infection abroad.

Add strain names to Figure 1 above the corresponding collection icons in the timeline.

Indicate antibiotic therapy durations graphically in Figure 1 (rather than putting a single point showing the initiation of antibiotics, put a range showing the start and stop time for the antibiotics. This will help give the readers a better understanding of the extent of the selection pressure.

Do the authors have any information on antibiotics that the patient may have received during his time in Vietnam or at Zhaongda hospital?

What was the patient's positive sputum culture in October and at the beginning of his third admission? Do the authors have these isolates available for analysis?

Were the patient's susceptible *K. pneumoniae* sputum and urinary cultures mentioned in line 81 the same ST? What was the susceptibility profile of these isolates? Is it the same as that of the blood culture isolate analyzed in this manuscript?

Authors must clarify their thoughts on the sequence of events and lay out the alternative explanations for the results. They can then say in the discussion how their results support or refute the different possibilities. A reader of the manuscript could draw the conclusion that the authors are saying that the antibiotic therapy of imipenem ceftazidime caused the evolution of a *K. pneumoniae* strain in the patient. The data in the paper does not support this claim. I would be more comfortable with the authors saying that they cannot determine the initial conditions (i.e. both clones of *K. pneumoniae* may have existed prior to the antimicrobial therapy and treatment with antibiotics may have conferred a selective advantage to the carbapenem resistant clone. Alternatively, the antibiotic therapy may have wiped out the susceptible *K. pneumoniae* and then the patient could have been infected by an entirely different clone through their frequent contact with the hospital environment.)

Lines 41-43: These lines do not make sense to me. Please rewrite for clarity. For example, I am not sure how the mutations showed great significance to prevent recurrent infection. I think the authors are trying to say the following several things at once. These statements need to be better supported by the text and results in the manuscript.

My interpretation of the authors claims in these lines:

1. Resolution of clinical symptoms after antibiotic therapy does not imply treatment success
2. Development of antibiotic resistance can lead to treatment failure and chronic infection
3. Mutations in *K. pneumoniae* collected longitudinally from a patient for this manuscript are an example of evolution of resistance as a response to antibiotic therapy

All three of these points need to be introduced more clearly in the abstract, importance, and introduction section. Points 1 and 2 need clear references attached to them. Currently point 1 is not mentioned at all in the introduction. Point 2 is only mentioned insofar as antibiotic resistance is a "global threat to patients and public health." The authors also need to lay out in the results and discussion section exactly how their results demonstrate point 3, because this is the crux of interest for the manuscript.

At the beginning of each results section, please include rationale for how the experiments contribute to proving the claims in introduction. "e.g. To show (subcomponent of what we are trying to prove) we did (X experiment)." Subsequently, in the discussion tie these results sections together to show that the experiments in the results prove the overall claims made in the manuscript.

A single growth curve comparing strains is not sufficient to say in the discussion "the spread of blaNDM-4 is of concern is no conspicuous fitness costs were identified in the self-conjugative plasmid containing blaNDM-4". This claim needs to be moderated or additional fitness cost experiments need to be provided.

Minor Comments

Comparing the presence or absence of antibiotic resistance genes, virulence factors, etc between isolates is difficult in the table format presented in Table 2. A presence or absence heatmap figure would be helpful in allowing readers to easily compare between strains.

Fix Table 1 so that IPM in the table and IMP in the legend are consistent.

Figure S4: please present data as individual points with the error bars instead of bar graphs.

Figure 2a: include gene annotation colors in legend

Line 57: reference required

Lines 59-60: reference required

Line 61-62: reference required

Lines 99-115: any configurable settings in any of the software used should be detailed in this section (i.e. what were cutoffs used for the different software to determine positive hits)

Lines 136-137: if any statistical analysis software was used, it should be listed here along with the commands used.

Line 188: Justification/precedent for 90% identity and 60% coverage cutoff is needed (e.g. default for the software, previously shown to have high accuracy, etc.)

Line 212: justification for identity cutoff

Lines 259-260: explicitly state in the text the significant differences wrt. Susceptibility

Line 290: reference required

Line 292-294: This sentence is not adequately proved in the manuscript

Line 312-313: reference required

Line 313-315: reference required, what is the relevance of this sentence to the manuscript findings?

There are syntax errors throughout the manuscript. I have detailed a few examples below, but I recommend that a copy editor go over the manuscript to fix other such errors.

Line 28: "were co-existed" remove were

Line 29: "contained" should be containing

Line 32: "which likely caused by" should be "which was likely caused by"

Line 38: spell out CRKP fully. (acronym not defined until the main manuscript)

Line 61: sequences play

Line 62: belonging

Line 65: define CPS

Line 70: "become more deeply"?

Line 120: amplified

Line 120: primers

Line 123: "11 hour" should be 11 hours

Line 136-137: and P-values <0.05 were considered significant

Line 147: retained

Line 152: considerable similarity

Line 184: remove "in size"

Line 196: No space between and lactamase

Line 202: suggesting homology extremely

Line 221: "resistant" should be resistance

Line 278-279: grammar for "this" vs "strains" either "this strain" or "these strains"

The authors are inconsistent with their use of beta spelled out and the Greek character . E.g. lines 292 and 298

Line 295: "surveillance" should be surveilling

Line 299: "patient discontinued imipenem" to "patient's imipenem therapy was discontinued" since the decision to discontinue the therapy was presumably made by the patient's medical team

Staff Comments:

Preparing Revision Guidelines

Please return the manuscript within 60 days; if you cannot complete the modification within this time period, please contact me. If you do not wish to modify the manuscript and prefer to submit it to another journal, please notify me of your decision immediately so that the manuscript may be formally withdrawn from consideration by Microbiology Spectrum.

Zhang et al., performed an interesting study about the identification and molecular characterization of CPE from the same patient. This study is very well written and clear. It is very interesting to identify the same *Klebsiella pneumoniae* clone with different genetic and virulence backgrounds. The authors showed that *K. pneumoniae* ST16 possessed a series adaptation in response to antibiotic pressure. Even so, there are some points the authors should consider before the acceptance of the manuscript.

-Abstract: It could be interesting to indicate the detected mutations in AcrR.

-Introduction: L56, the NDM-4 was identified from Egypt, not from Japan, please correct.

-L80-L81: how did you know *K. pneumoniae* was susceptible when recovered from sputum and urinary cultures during the first hospitalization of the patient? And on what basis the patient had received moxifloxacin for the treatment?

-L80-L87: If possible, include the dates of the patient admission and bacterial isolation on the text to prevent any conflict.

-On Antimicrobial susceptibility testing, you used EUCAST v.11.0 breakpoints for tigecycline, is the methodology of CLSI guidelines are the same as EUCAST? If not, it is better to repeat the experiment for tigecycline totally based on EUCAST guidelines?

-Please include a reference to the conjugation assay.

-L152-L54, blaOXA-1 is not carbapenemases-encoding genes, please exclude and correct this sentence. Also correct it in Table 2.

-Table 1, please indicate the meaning of bold and underlined values.

- Also, in table 1 please indicate (R), (I), or (S), after each value to indicate resistant, intermediate, or sensitive, respectively, and to prevent confusion.

- Please add the MIC results of quality control strains in Table 1.

-L257-259 is very confusing, please explain again why you can not get a transconjugant with blaOXA-181

-L258, please correct blaOXY-181.

- In the discussion, please discuss in detail if NDM-4 and/or OXA-181 producing *K. pneumoniae* ST16 have been previously identified or not? Which country? The possibility of their dissemination between countries?

-L293-L294, please explain your meaning with supporting reference(s).

-Several CRE isolates co-producing different carbapenemases and co-infection of with multiple CRE in the same patient has been identified, please discuss.

-Finally, what about the ethical approval of this study?

Dear Dr. Karen Carroll

Thank you for your letter and the reviewers' comments on our manuscript entitled "Repeated isolation of NDM-4 and OXA-181-Producing ST16 *Klebsiella pneumoniae* from a single patient in China" (ID : Spectrum02624-21). Those comments are very helpful for revising and improving our paper, as well as the important guiding significance to other researches. We have studied the comments carefully and made major revisions which we hope meet with approval. The main corrections are in the manuscript marked in yellow and the response to the reviewers' comments are as follows in bold:

Replies to reviewers' comments

Reviewer #1 (Comments for the Author):

Biying Zhang et al repeated isolation and identified four strains of ST16 *Klebsiella pneumoniae* isolates from the same hospitalized patient and succeeded in assessing the differences between strains by whole-genome analysis and phenotypic studies. The acquisition of *bla*_{NDM-4}-harbored or *bla*OXA-181-harbored plasmid and IS26-mediated HGT indicated the changes in the bacteria as they adapt to antibiotic treatment. The CZA-susceptible strain had a stronger ability in biofilm formation. The clinical background was sufficiently well described and multiple methods were used to elucidate that the phenotype and antibiotic resistance genes of the same clone were significantly diverse in different periods of antibiotic treatment.

The loss of some resistance genes revealed the bacterial adaptation strategies, which is beneficial to other groups studying the infectious prognosis and the source of persistent bacterial, not just those focused on the mechanism of bacterial resistance. The discussion linking the different analysis together was well explained. This study provides a reference for the in vivo evolution of clinical strains. Overall,

the paper is interesting, and data is well presented and well written.

Response: We sincerely appreciate this expert reviewer's accurate and objective assessment on our manuscript. In this study, we found that four samples isolated successively from a single patient all belonged to *Klebsiella pneumoniae* ST16, while showed differential susceptibility to ceftazidime/avibactam. By subsequent phylogenetic analysis, we confirmed that these strains could be assigned into two subpopulations of ST16 clone. According to the WGS data, both carried different resistance plasmids and had a risk of nosocomial transmission. In addition, by comparing differences between the same subpopulation, we identified gene transfer mediated by IS26, which implied the dynamics of the bacteria *in vivo* and emphasized the importance of increased surveillance during antibiotic treatment.

Minor revision:

1. The font size of the figure of the clinical background can be larger.

Response: Thanks for the suggestions. We have resized the font of Figure 1.

2. Please pay attention to the spelling of "conjugant" in Table 1.

Response: Thanks for the reviewer's reminder, We have made modifications in our revised manuscript.

3. Line 220: "Plasmid MLST (pMLST) analyses identified the same Incs of plasmids in K191724, K191725 and K191773..." According to Table 2, the Incs in K191724 was different from K191725 and K191773. Please revise this part.

Response: As the reviewer mentioned, we didn't detect the plasmid IncX3 and IncI-I(Alpha) replicon types in the draft genome sequence of K191724. However, we found the same resistance genes *bla*_{CMY-2} and *bla*_{OXA-181} in K191724, K191725, and K191773. According to the complete

genome sequence of K191773, *bla*_{CMY-2}-harboring plasmid belongs to IncI-I(Alpha), and *bla*_{OXA-181}-harboring plasmid belongs to ColKP3-IncX3. In addition, we designed primers for NikB, a IncI1-type relaxase, and the result of PCR confirmed the existence of this gene in K191724. It can be inferred that K191724 carried the same type of plasmids. For prudence, We have changed “same Incs” into “similar Incs” in the revised manuscript (Line 217).

4. Line 228: "IS26- aac(6')-Ib-cr-*bla*OXA-1- *cat*B3-IS26", the gene names here are inconsistent with the annotation in Figure 3.

Response: We are very sorry for our negligence that resulted in inconsistencies between images and annotations. After a careful check, we found that there was a problem with the genetic labeling in Figure 3. Thanks for your reminder, and We have changed “*cat*B” to “*cat*B3” in the revised version of Figure 3.

Reviewer #2 (Comments for the Author):

Overall, this study provided interesting data on the repeated isolation of carbapenemases bearing K. pneumoniae ST16 clinical samples in a patient in China. The authors performed antimicrobial susceptibility testing (AST), conjugation/biofilm assays and whole-genome sequencing (WGS). They also compared their plasmid sequences to other related samples in other countries. However, the manuscript was not well written with quite a few errors and several statements/data lacked strong evidence or support.

Response: Thanks to the reviewer for your comments concerning our manuscript. Those comments are all valuable and very helpful for revising and improving our paper, as well as the important guiding significance to our research.

We are very sorry that the importance of this study had not been explicitly stated, and some unwarranted conclusions had been drawn. The main findings of this study are as follow:

(1). The strain isolated from the bloodstream (K191663) was different from the other three NDM-4-producing ST16 *K. pneumoniae* strains (K191724, K191725, K191773). K191663 was susceptible to ceftazidime/avibactam (CZA) and carried more virulence factors. What's more, a frameshift mutation resulting from 1-bp deletion in AcrR of K191663 was detected. In previous reports, AcrR plays a pivotal role not only in drug resistance but also in virulence.

(2). The *bla*_{NDM-4}-harboring plasmid was the leading cause of resistance to carbapenems in K191724, K191725, and K191773. According to the reviewer's suggestions, We have performed conjugation assay, growth curves assay, plasmid stability, and competition assay to verify that this conjugative plasmid can transfer into recipient *E. coli* J53 with high conjugation frequency, although the biological cost was imposed on the recipient.

(3). Our patient came back from Vietnam, while the genome of K191724, K191725, K191773 were closely related to a strain isolated in Korea (18-2374). In addition, compared with K191724 and K191725, 12-kbp fragment possessing a multidrug-resistant region was lost mediated by IS26 in K191773, which changed the MICs of chloramphenicol slightly (MIC of K191724 and K191725 were 16 mg/L, K191773 was 8 mg/L), indicated the evolution of plasmid *in vivo*.

Overall, our study described two subpopulations of *K. pneumoniae* ST16 from a single patient during the antibiotic therapy and analyzed the characteristics of plasmids carried by each strain. Studying the diversity of strains within-patient is helpful to control the progression of diseases and necessary to understand the potential evolution of plasmids. We have carefully revised the manuscript according to the valuable comments of the reviewer.

1. In order to determine the origin of KP ST16 in China, could you do a genome comparison/phylogeny (chromosomal DNA) to the sample from Korea (line 184) and other parts of China (any additional ST16 in other cities)? Can you run Kleborate on your genome sequences to determine the potential virulence factors?

Response: (1) Thanks for the reviewers' comments which are all valuable and very helpful for revising and improving our paper. We have performed single-nucleotide polymorphism analysis and constructed a phylogenetic tree based on concatenated genome sequence data of 48 international ST16 *K. pneumoniae*, including those detected in China, Thailand, Vietnam, and Korea, according to the reviewer's comments. We were surprised to find a close relatedness between CZA-resistant strains and a Korean strain, whereas the CZA-susceptible was clustered with four strains associated with nosocomial flora in China.

(2) Kleborate is a professional software used in *K. pneumoniae* genome analysis. We have used this software to predict the virulence factors, and We have got the similar result with VFBD and Kleborate, so we did not make changes to Table 2. Nevertheless, some of the details about yersiniabactin (*ybt*) were added to the text, including the yersiniabactin sequence type, *ybt* locus, and ICE*Kp* of K191663 were YbST 174, *ybt9*, and ICE*Kp3*, respectively. What is more, the relevant discussions are also presented in the revised manuscript. (Result: Line 188-191, Discussion: Line 342-345)

2. It was interesting to observe the loss of 12-kbp fragment in K191773 after stop using imipenem. What happen when K191773 is grown on media with different conc. of imipenem again?

Response: Thanks for the reviewer's warm comments. The loss of 12-kbp fragment in K191773 was of concern since it may imply the presence of compensatory mutations in K191773 or the

presence of horizontal transfer of resistance genes mediated by IS26. Unfortunately, the patient was not continuously monitored by collecting specimens from different body sites.

In order to investigate the persister formation of the final isolate, K191773, We have performed a concentration-dependent kill curve and chose the final concentrations of antibiotics were of 10-, 50-, 100- or 200-times the MIC. According to the reference “Survival of bactericidal antibiotic treatment by tolerant persister cells of *Klebsiella pneumoniae*” (DOI: 10.1099/jmm.0.000680). Since the MIC of meropenem proposed by the reviewer was higher than 128 mg/L and not suitable for this experiment. Therefore, we chose polymyxin B, for its relatively low MIC (0.5 mg/L). Compared with the killing kinetics of *K. pneumoniae* ATCC 700603, higher doses of the antibiotic decreased survival of K191773 effectively. (Fig 5c)

3. Did the authors isolate other carbapenemase producing Enterobacterales from the same stool samples?

Response: Thanks for the comments. We isolated only one carbapenemase-producing *K. pneumoniae* from the same stool sample.

4. Fitness cost cannot be estimated based on growth kinetics (Fig. 5) alone. It would be better to perform competition assay between the parental and conjugated strains! What is the plasmid stability?

Response:(1) Thanks for the helpful suggestions, which are very helpful for revising and improving our paper. It is really true as the reviewer mentioned that the growth curve was limited to estimate the fitness cost of plasmids. Accordingly, *In vitro* competition assay and stability test were supplemented in a revised manuscript. Our data showed that pOXA1-191663 harbored by CZA-susceptible strains carried no conspicuous fitness cost, while pNDM4-191773 harbored by CZA-resistant strains had a statistically significant reduction of fitness *in vitro*.

(2) After ten days of successive passages in antibiotic-free broth, both pOXA1-191663 and pNDM4-191773 were stably maintained in the transconjugants hosts (98.67–99.33% fraction of plasmid-containing lineages). The results related to the above two experiments are shown in Figures 5b and 5d. In addition, we added the conjugation frequency of these plasmids, and results showed that both of them can transmit resistance genes at room temperature. (Line 304-308)

5. In the biofilm formation experiment, it is important to have controls! Otherwise, how to justify their abilities in colonization?

Response: We feel very sorry that we did not consider the control group for the biofilm experiment. We have chosen *K. pneumoniae* NTUH-K2044 as a control, which carries significant quantities of capsular polysaccharides for biofilm formation. With the help of the reviewer's suggestions, we found that K191663 had a stronger biofilm formation ability than K2044, while CZA-resistant strains' biofilm formation ability was weaker than K2044. However, the biofilm phenotype was not fit within the scope of this manuscript, and We have removed this result from our revised version.

Minor corrects/clarification:

Any written consent from the patient/ IRB for this study?

Response: Thanks for the warm comments. The clinical case involved in this study has obtained

ethical approval and informed consent of the patient. The ethical code is 20180309059. (Line 167)

The current title does not reflect the content/aims of the investigation, which appeared to illustrate the acquisition of plasmid in the maintenance of carbapenemase in KP ST16.

Response: Thanks for the insightful suggestions. It is really true as the reviewer mentioned that the current title does not reflect the aims of our study. We would like to change it to “Comparison of two distinct subpopulations of *Klebsiella pneumoniae* ST16 co-occur in a single patient”.

The abstract and "importance" will need some revisions to tighten the content. For example, line 26-27, this sentence was poorly written.

Response: Thanks for the insightful suggestions. We made extensive changes to the Abstract, Importance and Introduction. The poorly written sentences mentioned by the reviewer were removed. We hope the reviewers will be satisfied with the revised content.

L69, add *K. pneumoniae* after clinical

Response: Thanks for the helpful suggestions. Since the original sentence “Understanding the mechanism of...evolution of clinical strains.” was pointed out to be inconsistent with the claims of this study, we revised this sentence and added *K. pneumoniae* after the clinical according to the reviewer’s suggestions. (Line 75-76)

L70, "deeply" means important/ significant?

Response: Thanks for the helpful suggestions. We feel very sorry for the incorrect descriptions in our manuscript. We have revised the Introduction, and removed those incorrect descriptions.

L72, replace "reasons" with mechanisms; replace "reference" by "baseline information"

Response: Thanks for the helpful suggestions. We have rephrased the sentence according to reviewer’s suggestions. (Line 34)

L87, What kind of "symptomatic treatment"?

Response: The patient has some underlying disease of the heart function and therefore received treatment for cardiac insufficiency, chest distress and asthma.

L96, Why not a *Klebsiella pneumoniae* was used as a reference? *K. quasipneumoniae* is a different species.

Response: Thanks for the warm comments. As previous report described, *Klebsiella quasipneumoniae* subsp. *similipneumoniae* strain ATCC 700603, formerly known as *K. pneumoniae* K6 and was reclassified as a new species in recent years. We feel sorry for any misunderstandings caused to the reviewer and we have changed this part to *K. pneumoniae* ATCC 700603. (Line 101-102)

L97, Which CLSI session?

Response: Thanks for the comments. The interpretation of the MIC results was based on 2020 CLSI M100 30th Edition breakpoints. We have refined it in the revised manuscript. (Line 103)

L 101, remove "additionally"

Response: Thanks for the helpful suggestions. We have removed this word in the revised manuscript. (Line 107)

L115, add reference for MEGA-X

Response: Thanks for the warm suggestions. We have added a reference for MEGA X. (Line 125)

L120, replace "amplificated" with "amplified"

Response: We feel sorry for our careless writing and thanks for the warm suggestions. We have revised the error here. (Line 130)

L136, which software was used to run ANOVA?

Response: Thanks for the comments which help us to improve our manuscript. The ANOVA analyses were calculated using Prism7 (GraphPad Software). The original sentence was modified since the biofilm formation test was deleted. We used the Student's t-test with GraphPad Prism software 7.0 in the subsequent statistical analysis of conjugation frequency. (Line 158)

L139-142, please add SRA #, BioProject #

Response: Thanks for the comments. Raw sequence reads are available on NCBI under Bioproject accession number PRJNA749815. (Line 164-165)

Table 1. J53, J63, J73 should be listed at the bottom of the table.

Response: Thanks for the insightful suggestions. We have rearranged the layout of Table 1 according to the reviewer's suggestion.

Fig.2. Diagram 2a was generated using BRIG. Which software was used to generate diagram 2b?

Response: Thanks for the comments. We downloaded the data in GeneBank format from NCBI, and then opened them with SnapGene. By carefully comparing the size and direction of genes, we drew Diagram 4b with PowerPoint. (We have changed the order of the diagrams)

Fig S3. Better to plot the tree as a phylogram and put the bootstrap values on the branch. Difficult to interpret the relationship of these samples.

Response: Thanks for the insightful suggestions. As the reviewer mentioned, FigS3 failed to clearly show the relatedness of all the samples. We have recreated a phylogenetic neighbor-joining tree and labeled the bootstrap on the branches.

L280, change "multi-resistance" to multi-drug resistance

Response: Thanks for the warm suggestions. We have changed the statement in the revised version. (Line 367)

L283, there was no strong evidence from the investigation to support "increase persistence over long periods"

Response: Thanks for the comments. It is really true as the reviewer mentioned that we couldn't provide evidence that bacteria maintain their survival in the host by acquiring resistant plasmids.

In this study, it is likely that positive selections led to the repeated isolation of multi-drug resistant *K. pneumoniae* ST16. We have removed this sentence and the Discussion has been modified.

L293, please rephrase "..can last longer under pressure .."

Response: Thanks for the comments. We feel very sorry for the incorrect description. In fact, we could not demonstrate that the *bla*_{NDM-4}-harboring plasmid can be stably present in bacteria in the presence of different antibiotics. Thus, this sentence was deleted in revised version.

L294, rephrase " .. we reported, for the first time the coexistence.." as "this is the first report on the coexistence .."

Response: Thanks for the helpful suggestions. We have revised the sentence according to reviewer suggestions. (Line 381-382)

L309, "free fitness cost" was bad description

Response: We feel very sorry for the bad description about the fitness cost carried by plasmids.

According to the results of competition assays, pOXA1-191663 produced non-inferior biological cost. Consequently, We have redescribed this sentence. (Line 373-375)

L310, what kind of virulence factors?

Response: Thanks for the warm comments. The mechanism of biofilm formation among the clinical strains of *K. pneumoniae* is reported to be associated with a plethora of genes. Few

significant genes include aerobactin (*iutA*), type I fimbriae (*fimA* and *fimH*), type III fimbriae (*mrkA* and *mrkD*), capsular polysaccharide (CPS). In addition, the yersiniabactin uptake receptor *fyuA* is also required for biofilm formation. Given the quorum sensing also affects biofilm formation, it is unreasonable for us to explain the stronger biofilm formation of K191663 in this way. Although we have removed the biofilm-related content, we thank the reviewer for constructive suggestions.

L317, this is not a study on the evolution of ST16 in China.

Response: Thanks for the insightful comments which are the important guiding significance to our research. We found that the four strains isolated in this study belong to two subpopulations of *K. pneumoniae* ST16 through phylogenetic analyses. As a result, the connection between strains cannot be defined as evolution. We have deleted this sentence and changed the description. However, we believe that the loss of 12-kbp fragments belongs to the evolutionary category. Because IS26 has been previously reported to mediate the loss of resistance gene, and our plasmid comparison results also showed that the surrounding structures of two plasmids with the same replicons were identical except for the multi-drug-resistant region.

Reviewer #3 (Comments for the Author):

The authors perform characterization and whole-genome sequencing of 4 *Klebsiella pneumoniae* isolates obtained from a hospitalized patient in China. The isolates undergo Illumina and PacBio sequencing, antibiotic susceptibility minimum inhibitory concentration testing, and biofilm characterization. The genomic assembly, annotation, and characterization identifies molecular determinants for carbapenem resistance (*bla*NDM, and *bla*OXA-181 mediated). Further, the use of

conjugation assays with *E. coli* J53 demonstrate that carbapenem resistance is transferable. The plasmid and resistance gene analysis is impressive.

However, the genomic characterization described is insufficient to classify the relatedness of the isolates. The authors use of ANI for this purpose is not suitable compared to the alternative (whole-genome alignment or core-genome alignment). Accordingly, it is not exactly clear whether this is persistence of the same clone or coexistence of two clones within the same sequence type. The biofilm characterization does not fit within the scope of this manuscript and should be removed for consideration.

Response: Thanks to the reviewer for your comments concerning our manuscript. Those comments are all valuable and very helpful for revising and improving our paper.

(1). It is true as the reviewer commented that this manuscript had not clearly described the relatedness among these four isolates. Accordingly, we performed single-nucleotide polymorphism analysis and constructed a phylogenetic tree based on concatenated genome sequence data of 48 international ST16 *K. pneumoniae*. The results confirmed the reviewers' comments that there was a coexistence of two subpopulations of ST16 in the same patient. It was the combined treatment of CZA and imipenem that resulted in the CZA-resistant strains becoming the dominant bacteria in the patient.

(2). Regarding biofilm formation, it may not fit the claims of this study, as the reviewer said.

Although there was a significant difference in biofilm formation ability between CZA-susceptible and CZA-resistant strains, we concluded that these differences did not result from antibiotics treatment through phylogenetic analysis. Therefore, we removed it according to the reviewer's suggestions.

Minor Comments

Line 65 - Define CPS (and all acronyms at first use)

Response: Thanks for the kind suggestion. The original sentence has been removed because We have made extensive changes to the Introduction. However, We have carefully checked the manuscript for all acronyms at first use to ensure that they were defined accordingly.

Line 81 - Susceptible to what? Be specific

Response: Thanks for the comments. It was carbapenems-susceptible *K. pneumoniae* isolated from sputum and urinary. We have revised in the manuscript. (Line 85)

Line 120 - Do you mean primers instead of "premiers"?

Response: We feel very sorry for this misspelling and We have corrected it in the manuscript. (Line 130)

Line 127 - Were the plates TC treated or not?

Response: Thanks for the warm comments. The tissue culture plate has not been treated.

Lines 139-142 - I am able to see that the strains are deposited in NCBI. I would also recommend that the authors reference their assigned BioProject ID so that people interested in their strains have a single webpage to see what they sequenced.

Response: Thanks for the kind suggestions. The BioProject ID has been added into the Data Availability. (Line 164-165)

Major Comments

Lines 151-152 & Line 181 -182 - In this context, ANI really should just be used to verify that the isolates are in fact *Klebsiella pneumoniae* (and not another member of the species complex). I would

not support this statement without additional SNP analysis to verify clonality. I would recommend that the authors remove this from the text or perform more SNP analysis.

Response: We are very sorry about the inappropriate use of ANI analysis in this study. We have performed SNP analysis and constructed a phylogenetic tree to clarify the relatedness among the four strains. The result indicated that the three NDM-4-producing strains were more similar, while K191663, isolated from blood, was a different clone of ST16 *K. pneumoniae*. Thanks to reviewer for the helpful comments, which help us analyze the genome more accurately. (Figure 2)

Lines 307-308 - This is a major overstatement. The study did not examine therapy outcomes in single versus combination therapy. This study examined Klebsiella isolates.

Response: We feel very sorry for the overstatement about clinical therapy. According to the case, the patient had been treated with imipenem for urinary tract infection before K191663 was isolated. However, a bloodstream infection developed and imipenem-resistant K191663 was isolated. So, the CZA was added for combination therapy, which proved to be effective. It is really true as the reviewer mentioned that the purpose of this study is not to compare the outcomes of monotherapy and combination therapy. We have removed this sentence in the revised manuscript.

Lines 310-311 - This is too speculative, even for a discussion. Biofilm formation is a multifactorial process and it is difficult to attribute individual gene presence/absence to this phenotype.

Response: We feel very sorry for attributing reduced biofilm formation to the absence of virulence factors solely. In addition to the expression of biofilm-related genes, quorum sensing, efflux pump, and cyclic dinucleotide signaling pathway also affect the biofilm formation. Given

the biofilm characteristic was not fit within the scope of this manuscript, We have removed the biofilm-related content, and we will avoid such mistakes in further studies.

Lines 312-313 - Please cite literature indicating that the AcrAB gene components are important for biofilm formation.

Response: Thanks for the reviewer's comments. According to the article “Secondary multidrug efflux pump mutants alter Escherichia coli biofilm growth in the presence of cationic antimicrobial compounds” (DOI: 10.1016/j.resmic.2016.11.003), *E. coli* mutant strains lacking the efflux genes *acrB*, *acrE*, and *tolC* displayed significant reductions in biofilm growth. Given that the biofilm characterization has been removed, we did not present the connection between AcrAB-TolC efflux pump and biofilm in the revised manuscript.

Reviewer #4 (Comments for the Author):

Zhang et al., performed an interesting study about the identification and molecular characterization of CPE from the same patient. This study is very well written and clear. It is very interesting to identify the same *Klebsiella pneumoniae* clone with different genetic and virulence backgrounds. The authors showed that *K. pneumoniae* ST16 possessed a series adaptation in response to antibiotic pressure. Even so, there are some points the authors should consider before the acceptance of the manuscript.

Response: Thanks for the reviewers' comments which are all valuable and very helpful for revising and improving our paper. Further analysis of the genome revealed that although our patient came from Vietnam, where there have been reported nosocomial outbreaks associated with *bla*_{NDM-4} or *bla*_{OXA-181}-producing *Klebsiella pneumoniae* ST16, the strains co-producing

NDM-4 and OXA-181 in this study were closely related to a strain reported in Korea.

-Abstract: It could be interesting to indicate the detected mutations in *AcrR*.

Response: Thanks for the insightful suggestions. The mutations in *acrR* of K191663 have been added to the Abstract. (Line 33-34)

-Introduction: L56, the NDM-4 was identified from Egypt, not from Japan, please correct.

Response: Thanks for pointing this out. At the same time, we feel very sorry for the mistakes here. The “Japan” has been corrected to “Egypt” in the revised manuscript. (Line 55)

-L80-L81: how did you know *K. pneumoniae* was susceptible when recovered from sputum and urinary cultures during the first hospitalization of the patient? And on what basis the patient had received moxifloxacin for the treatment?

Response: Thanks for the insightful comments. (1). In this study, we isolated four carbapenems-resistant strains, all belonging to ST16 *K. pneumoniae*, but they differed in susceptibility to CZA, which stimulated our interest in exploring them in depth. We did antimicrobial susceptibility testing and confirmed that the strains isolated from sputum and urinary cultures were susceptible to carbapenems, as well as to a variety of β -lactam (ceftazidime, cefotaxime). To avoid confusion, we classified them as carbapenem-susceptible strains.

(2). Regarding the question "On what basis the patient had received moxifloxacin for the treatment?" Given that moxifloxacin belongs to the quinolone class and is mainly used clinically to treat adults with upper and lower respiratory tract infections. Our patient had a tracheotomy during his hospitalization at Zhongda hospital and the bacteria isolated from the urinary tract were antibiotics-susceptible, so moxifloxacin was chosen for anti-infective treatment.

-L80-L87: If possible, include the dates of the patient admission and bacterial isolation on the text to prevent any conflict.

Response: Thanks for the warm suggestions. We have revised the case report according to the reviewer's comments and added the dates of patient admission and bacterial isolation on the text. At the same time, We have labeled the dates of antibiotic use in Figure 1 to show the connection between resistant strains and antibiotic treatment.

-On Antimicrobial susceptibility testing, you used EUCAST v.11.0 breakpoints for tigecycline, is the methodology of CLSI guidelines are the same as EUCAST? If not, it is better to repeat the experiment for tigecycline totally based on EUCAST guidelines?

Response: Thanks for the comments. The EUCAST recommendations for MIC determination for non-fastidious organisms are in complete agreement with the recommendations from the International Standards Organization standard 20776-1. Accordingly, we checked the ISO 20776-1:2019 and found that the broth micro-dilution method described in the document is identical to the CLSI. In summary, the methodology of CLSI guidelines is the same as EUCAST.

-Please include a reference to the conjugation assay.

Response: Thanks for the reviewer's helpful suggestions. We have provided references for conjugation assay. (Line 128)

-L152-L54, blaOXA-1 is not carbapenemases-encoding genes, please exclude and correct this sentence.

Also correct it in Table 2.

Response: We feel very sorry for our careless writing. As the reviewer's comments, *bla*_{OXA-1} is a resistant gene encoding a β -lactamase. We have corrected this sentence and relevant contents in Table 2.

-Table 1, please indicate the meaning of bold and underlined values.

Response: Thanks for the reviewer's warm suggestions. The sentence "Bold flags susceptible, and underlined represents intermediate." had been marked in the comments at the bottom of the Table 1. (Line 198-199)

- Also, in table 1 please indicate (R), (I), or (S), after each value to indicate resistant, intermediate, or sensitive, respectively, and to prevent confusion.

Response: Thanks for the reviewer's warm suggestions. Since there was not enough space to label all the results with (R), (I), or (S) in Table I, we use bold and underline to represent susceptible and intermediate, respectively. In addition, these non-resistant results are also labeled in the upper right corner with S or I

- Please add the MIC results of quality control strains in Table 1.

Response: We feel very sorry for not showing the MICs of the quality control strains in the Table 1, which may have caused some confusion for reviewer and readers. We have added the MIC results of *E. coli* ATCC 25922 and *K. pneumoniae* ATCC 700603 in the last two rows of Table 1.

L257-259 is very confusing, please explain again why you cannot get a transconjugant with

*bla*OXA-181

Response: Thanks for the warm comments. In fact, *bla*_{OXA-181}-harbored plasmid(pOXA181-191773) was conjugative. However, under the selection of 0.3 mg/L meropenem, we did not detect transconjugants only carrying pOXA181-191773; instead, among the transconjugants containing this plasmid, we also detected the *bla*_{NDM-4}-harbored plasmids(pNDM4-191773). We speculated that the transfer of this plasmid requires other self-conjugative plasmids as a helper plasmid (for example, the pNDM4-191773).

-L258, please correct blaOXY-181.

Response: Thanks for the warm suggestions. We have corrected this part in the revised version.

(Line 264)

- In the discussion, please discuss in detail if NDM-4 and/or OXA-181 producing *K. pneumoniae* ST16 have been previously identified or not? Which country? The possibility of their dissemination between countries?

Response: Thanks for the warm suggestions. According to previous reports, Nosocomial outbreaks caused by *K. pneumoniae* ST16 carrying *bla*_{NDM-4} or *bla*_{OXA-181} have been reported in Vietnam, where our patient returned. However, they did not have a significant correlation with our CZA-resistant isolates. In contrast, the coexistence of *bla*_{NDM-4} and *bla*_{OXA-181} in *K. pneumoniae* ST16 was only described in South Korea and showed a close relatedness with our isolates. Given that our patient has not been to South Korea, it is unknown how they became infected with a nearly similar clone. Regarding the issues raised by the reviewer, we have added them in the Discussion section. (Line 346-351)

-L293-L294, please explain your meaning with supporting reference(s).

Response: We feel very sorry for the incorrect description. In fact, we could not demonstrate that the *bla*_{NDM-4}-harbored plasmid can be stably present in bacteria in the presence of different antibiotics. Thus, this sentence was deleted in revised version.

-Several CRE isolates co-producing different carbapenemases and co-infection of with multiple CRE in the same patient has been identified, please discuss.

Response: Thanks for the helpful suggestions. It is really true as the reviewer said that the coexistence of two classes of carbapenemases has been detected globally and could intensify substantial challenges for treatment. We have reported that KPC-2 and NDM-5 co-existed CRKP ST11 in a patient last year. These researches of two carbapenem-resistant genes carried by the same strain increase the awareness of carbapenem-resistant Enterobacteriaceae. We have added related discussion in the revised manuscript. (Line 378-383)

-Finally, what about the ethical approval of this study?

Response: Thanks for the warm comments. The clinical case involved in this study has obtained ethical approval and informed consent of the patient. The ethical code is 20180309059. (Line 167)

Reviewer #6 (Comments for the Author):

Repeated isolation of NDM-4 and OXA-181-Producing ST16 *Klebsiella pneumoniae* strain from a single patient in China

In this manuscript, the authors analyze antibiotic susceptibility, antibiotic resistance genes, virulence genes, and mobile genetic elements in four *K. pneumoniae* clinical isolates. These isolates were taken longitudinally from a 45-year individual with multiple comorbidities which included urinary tract

infection, culture positive blood stream infection, and culture positive pneumonia. The patient's treatment included multiple antibiotic courses with differing susceptibility results from the patient bacterial isolates.

The authors should be commended for presenting an interesting genomics study grounded in a clinically interesting case. Beta-lactams are some of the most used antibiotics in clinical settings and *K. pneumoniae* is a common hospital pathogen. The spread of carbapenem resistance should concern all clinicians worldwide. The authors should also be commended for laying out how some of their findings fit with findings from other groups. Unfortunately, in its present form, the sections of the manuscript do not fit together to form a cohesive whole.

Response: Thanks to the reviewer for valuable time and detailed and constructive comments on our manuscript. As the reviewer said, our manuscript did overlook many details, the “Result” and “Discussion” were not combined to highlight our claims. In this study, our study aims to elucidate the diversity of ST16 *K. pneumoniae* through comparative analysis of whole-genome sequencing data and identify the potential evolution of plasmids by sequencing longitudinal clinical isolates during antibiotic treatment. Therefore, we used phylogenetic analysis to determine the origin of pathogenic bacteria. Whole-genome sequencing was performed to predict resistance genes and virulence factors and to obtain mutations in both genomes and plasmids. In addition, we combined antimicrobial susceptibility testing to analyze the cause of antibiotic resistance of bacteria. Conjugation and stability tests were used to indicate the horizontal transfer of resistance genes with plasmids.

We have studied these comments carefully and tried our best to revise and make great changes to the manuscript. We hope the correction will meet with approval.

1. The overarching claims of the manuscript are not succinctly laid out beyond that antibiotic resistant *K. pneumoniae* isolates were collected from a patient in China.

Response: We are very sorry for neglecting to clearly articulate the claims of this study in the manuscript. In addition to isolating CRKP in the same patient, our primary aim was to describe the diversity between these clinical cultures and the causes of antibiotic resistance. Thanks for the helpful comments, which help us improve our manuscript.

2. The Introduction does not provide adequate background for the claims made in the paper or set the reader up to understand where the authors are trying to lead with the subsequent Results and Discussion sections. Some sections of the Introduction are extraneous and critical pieces of information for understanding the background for the manuscript are missing. Other sections of the Introduction lead the reader to expect results or findings that are not addressed in the manuscript.

Response: Thanks for the insightful comments which helps us find the weaknesses in the "Introduction" and improve our manuscript. Firstly, we feel very sorry for the insufficient background in the Introduction section and any burden that may have been caused to the reviewer's understanding of this paper. In the revised version, we added the prevalence of *K. pneumoniae* ST16 in Vietnam and China, respectively, and the background on *acrR* mutations (Line 58-62, 71-74). Secondly, we removed the extraneous parts, including the differences between NDM-4 and NDM-1 mentioned by the reviewer below, and two sentences in lines 65-68

of the original paper, because we felt they were inconsistent with the claims of the revised manuscript and were unable to lead the readers to the subsequent results and discussions.

3. The Results section does not have strong descriptions of why the authors conducted the experiments, or how said experiments fit together to prove the overarching claims of the manuscript.

Response: It is really true as the reviewer commented that the relationship between the experiments and the claims is not declared in the “Result” section, which may confuse the readers with the purpose of the experiments. Thanks for the warm suggestions. We have added sentences about the purpose of the experiments in “Result”. In addition, we have rewritten the Discussion to combine the results. (Line 215-216, 236, 282-283, 304, 310-311, 325-326)

4. The Discussion does not flow from section to section, and it does not put the results into cohesive narrative. It also does not discuss the limitations of the study.

Though the manuscript contents are of interest to the readership of Microbiology Spectrum, this work requires major revisions.

Response: We feel very compunctious for the incorrect writing of the “Discussion” section. As the reviewer said, although we tried our best to discuss each result, we did not combine them together to prove the point of this study. We have rewritten this part in the revised manuscript. We thought the limitations of this study were that it did not continue monitoring and made us unable to know whether the lost fragment of plasmids is recombined again and whether its occurrence is related to specific anatomical sites. We have added the limitations in the Discussion (Line 364-365). Thanks again for the reviewer’s affirmation of our research and patient teaching.

Major Comments

The introductory paragraph needs to be rewritten to focus more on providing background for the scientific questions answered by the manuscript. For example, is the information on amino acid substitution for NDM-1 to NDM-4 relevant to the claims of this manuscript? In the introduction, the authors state, "Understanding the mechanism of resistance gene acquisition and stability could enhance our understanding of the evolution of clinical strains." However, they do not address the mechanism of gene acquisition or stability in the manuscript.

Response: We are grateful to the reviewer for pointing out our shortcomings in detail. These comments are of great help to us. When analyzing the genetic surrounding of bla_{NDM-4} , we found that it was very similar to bla_{NDM-1} in pNDM1-45001, carried by an *E. xiangfangensis* isolated in China. The differences between them included the opposite direction of the genetic surrounding and the substitution of amino acids within the bla_{NDM} . Consequently, the relationship between NDM-1 and NDM-4 was mentioned in the Introduction. We have carefully considered the comments of the reviewer, and it is true that unimportant or exaggerated information will cause confusion to readers. Therefore, the Introduction has been rewritten and the sentence "Understanding the mechanism..." has been removed.

After finishing the introduction, the reader should be able to anticipate what results will be shown in the following sections. For example, in the Biofilm section of the results, the authors say, "Biofilm experiment of isolates were carried out to investigate their ability to colonization." However, biofilms and colonization are not mentioned in the introduction and no references are given. Try to tie in the

different results sections to the introduction and provide the basic rationale for conducting the analysis presented in the manuscript.

Response: Thanks for the constructive comments. It is true as the reviewer's comments that the Introduction failed to tie in the result to the background. We have rewritten the Introduction based on the reviewers' guidance and ensured that the rewritten content could be integrated with the Result and Discussion.

In the discussion, the authors say that they report "for the first time the coexistence of blaNDM-4 and blaOXA-181 in ST16 *K. pneumoniae* in China, to highlight the importance of surveillance [sic] the prevalence of NDM enzymes and its latent vectors." However, *K. pneumoniae* sequence types and their global distribution are not touched on in the Introduction. The authors should introduce sequence types and why the reader should care about them in the Introduction. Specifically, they should provide any background for the reader to understand why they should care that ST16 is now in China and where ST16 is currently prevalent.

Response: Thanks for the helpful suggestions. In this study, our patient returned from Vietnam, where *bla*_{NDM-4}-carrying *K. pneumoniae* ST16 has been reported. In China, ST16-related nosocomial infections have been reported too, which requires more attention. According to the reviewer's suggestions, we have added the background about ST16 *K. pneumoniae* in Vietnam and China. (Line 60-62)

Can the authors comment on the prevalence of Clone ST16 in Vietnam? The paper seems to suggest that the individual acquired the infection abroad.

Response: Thanks for the insightful suggestions. So far, there have been two reports in Vietnam on the isolation of *K. pneumoniae* ST16 from inpatients. The strains in these two studies carried *bla*_{NDM-4} and *bla*_{OXA-181}, respectively. Unexpectedly, these strains in Vietnam differed from the isolates in our study according to the genome phylogeny analysis. The background of ST16 in Vietnam has been added in the Introduction, while the relationship among these strains was described in the Discussion. (Line 60-62; 346-348)

Add strain names to Figure 1 above the corresponding collection icons in the timeline.

Response: Thanks for the insightful suggestions. We have added strains' names in Figure 1.

Indicate antibiotic therapy durations graphically in Figure 1 (rather than putting a single point showing the initiation of antibiotics, put a range showing the start and stop time for the antibiotics. This will help give the readers a better understanding of the extent of the selection pressure.

Response: Thanks for the warm suggestions. We have added the antibiotic therapy durations both in Figure 1 and text.

Do the authors have any information on antibiotics that the patient may have received during his time in Vietnam or at Zhaongda hospital?

Response: Thanks for the helpful suggestions. However, we feel very sorry that we failed to find out the antibiotics the patient had used in Vietnam or Zhongda Hospital.

What was the patient's positive sputum culture in October and at the beginning of his third admission?

Do the authors have these isolates available for analysis?

Response: Thanks for the comments. The bacteria isolated in October was a strain of *K. pneumoniae* which belonged to ST656, and it was susceptible to carbapenems. Further genome

analysis revealed that no apparent connection between ST656 and ST16 was detected, so we did not include it in the manuscript. Unfortunately, we could not obtain the cultures isolated at the beginning of his third admission.

Were the patient's susceptible *K. pneumoniae* sputum and urinary cultures mentioned in line 81 the same ST? What was the susceptibility profile of these isolates? Is it the same as that of the blood culture isolate analyzed in this manuscript?

Response: Thanks for the warm comments. The susceptible *K. pneumoniae* strain (K191659) isolated from sputum belongs to ST656. It was susceptible to ceftazidime, ceftazidime-avibactam, cefotaxime, imipenem, meropenem, polymyxin B, and tigecycline. There was no obvious connection between K191659 and four ST16 strains. Genes encoding carbapenemases and β -lactamases were not detected in K191659. The WGS data of K191659 were deposited in the GenBank database with the BioSample number was SAMN20422816. Unfortunately, we could not obtain the cultures isolated from the urine for further analysis.

Authors must clarify their thoughts on the sequence of events and lay out the alternative explanations for the results. They can then say in the discussion how their results support or refute the different possibilities. A reader of the manuscript could draw the conclusion that the authors are saying that the antibiotic therapy of imipenem ceftazidime caused the evolution of a *K. pneumoniae* strain in the patient. The data in the paper does not support this claim. I would be more comfortable with the authors saying that they cannot determine the initial conditions (i.e. both clones of *K. pneumoniae* may have existed prior to the antimicrobial therapy and treatment with antibiotics may have conferred a selective advantage to the carbapenem resistant clone. Alternatively, the antibiotic therapy may have wiped out

the susceptible *K. pneumoniae* and then the patient could have been infected by an entirely different clone through their frequent contact with the hospital environment.)

Response: Thanks for the constructive suggestions, which help us to improve our manuscript. We have studied these comments carefully and made extensive revisions to the manuscript. As the comments by the reviewer, through phylogenetic analysis, we realized that those *K. pneumoniae* ST16 strains isolated before and after combination treatment with CZA and imipenem belonged to different subpopulations. It was the selection of antibiotics that caused CZA-resistant bacteria to become the dominant microflora, whereas the differences between the subsequent NDM-4-producing isolates may be related to the bacterial evolution *in vivo*. The main manifestation is IS26-mediated fragment loss. According to the previous report, following transposition events of mobile elements such as IS26 not only helps to understand the evolution of resistance plasmids, but also reveals the changes that may optimize the viability of the pathogen in a hospital environment.

Lines 41-43: These lines do not make sense to me. Please rewrite for clarity. For example, I am not sure how the mutations showed great significance to prevent recurrent infection. I think the authors are trying to say the following several things at once. These statements need to be better supported by the text and results in the manuscript.

My interpretation of the authors claims in these lines:

1. Resolution of clinical symptoms after antibiotic therapy does not imply treatment success
2. Development of antibiotic resistance can lead to treatment failure and chronic infection
3. Mutations in *K. pneumoniae* collected longitudinally from a patient for this manuscript are an example of evolution of resistance as a response to antibiotic therapy

All three of these points need to be introduced more clearly in the abstract, importance, and introduction section. Points 1 and 2 need clear references attached to them. Currently point 1 is not mentioned at all in the introduction. Point 2 is only mentioned insofar as antibiotic resistance is a "global threat to patients and public health." The authors also need to lay out in the results and discussion section exactly how their results demonstrate point 3, because this is the crux of interest for the manuscript.

Response: We feel very sorry for our poor writing, which has caused some confusion for the reviewer. (1). Regarding what we want to say in the Importance, we will explain as follows:

Pathogen can acquire antibiotic resistance by horizontal gene transfer, like resistant plasmids, or integration events by mobile elements. What is more, parts of bacteria can evolve into persistent cells, which are the major cause of recurrent infection after a successful course of therapy with antibiotics. We have described this background information in the Introduction and rewritten the Importance. (Line 65-66)

(2). The main findings of our study will be described as follows: First, we confirmed that *bla*_{NDM-4}-carrying plasmid was the major cause of drug resistance in *K. pneumoniae* ST16 through conjugant assay, antimicrobial susceptibility testing, and resistance genes analysis. Second, we identified IS26-mediated fragment loss leading to the emergence of a new plasmid without any resistance genes, through genomic comparisons and phylogenetic analysis. All of these results are based on continuous surveillance, as it helps us understand the occurrence and development of bacterial antibiotic resistance. Thanks for the warm comments, We have rewritten the Importance and Discussion in the revised version.

At the beginning of each results section, please include rationale for how the experiments contribute to proving the claims in introduction. "e.g. To show (subcomponent of what we are trying to prove) we did (X experiment)." Subsequently, in the discussion tie these results sections together to show that the experiments in the results prove the overall claims made in the manuscript.

Response: Thanks for the helpful suggestions. We have added the purpose of this experiment or analysis before describing the results according to the reviewer's suggestions. (Line 215-216, 236, 282-283, 304, 310-311, 325-326). In addition, we have rewritten the Discussion to combine the results.

A single growth curve comparing strains is not sufficient to say in the discussion "the spread of blaNDM-4 is of concern is no conspicuous fitness costs were identified in the self-conjugative plasmid containing blaNDM-4". This claim needs to be moderated or additional fitness cost experiments need to be provided.

Response: Thanks for the helpful suggestions which are very helpful for revising and improving our paper. It is really true as the reviewer mentioned that the growth curve was limited to estimate the fitness cost of plasmids. *In vitro* competition assay and stability test were supplemented in a revised manuscript (Fig 5b, 5d). Our data showed that pOXA1-191663 carried by CZA-susceptible strains carried no conspicuous fitness cost, while pNDM4-191773 carried by CZA-resistant strain had a statistically significant reduction of fitness *in vitro*. We have corrected the wrong description in the Result and Discussion. (Line 309-315, 374-376)

Minor Comments

Comparing the presence or absence of antibiotic resistance genes, virulence factors, etc between isolates is difficult in the table format presented in Table 2. A presence or absence heatmap figure would be helpful in allowing readers to easily compare between strains.

Response: Thanks for the kind suggestions. We have shown the resistance genes in a heatmap in Figure 2. However, we want to match the resistance genes to the related plasmids, and Table 2 was retained.

Fix Table 1 so that IPM in the table and IMP in the legend are consistent.

Response: Thanks for the kind suggestions. We have written the abbreviation of Imipenem as IPM uniformly. (Line 197)

Figure S4: please present data as individual points with the error bars instead of bar graphs.

Response: Thanks for the kind suggestions. The biofilm formation test has been deleted because it did not fit within the scope of this manuscript. The competition assay results have been present as individual points with the error bars according to the reviewer's suggestions. (Fig 5b)

Figure 2a: include gene annotation colors in legend

Response: Thanks for the kind suggestions. The gene annotations were written in-text annotation 6. (Line 256-257)

Line 57: reference required

Response: Thanks for the kind suggestions. We have added the references in regard to NDM-4 detected in China. (Line 56)

Lines 59-60: reference required

Response: Thanks for the comments. We have added the reference according to the reviewer's suggestions. (Line 59)

Line 61-62: reference required

Response: Thanks for the comments. We have modified the background of the insertion sequences. It has been reported that the bacterial insertion sequences strongly correlate with the evolution of resistance. We have added the reference after this sentence. (Line 67)

plasmids
Lines 99-115: any configurable settings in any of the software used should be detailed in this section (i.e. what were cutoffs used for the different software to determine positive hits)

Response: Thanks for the comments. We have added the references of SOAPdenovo v.2.04 and Unicycler v0.4.6. (Line 108-109) The parameters used by the online analysis site are system defaults, except the PlasmidFinder v.2.1, with the threshold for minimum 90% identity and minimum 60% coverage. (Line 114, 118)

Lines 136-137: if any statistical analysis software was used, it should be listed here along with the commands used.

Response: Thanks for the suggestions. We used Student's t-test with GraphPad Prism software 7.0 to compare the differences in transfer frequency and relative fitness. (Line 158-159)

Line 188: Justification/precedent for 90% identity and 60% coverage cutoff is needed (e.g. default for the software, previously shown to have high accuracy, etc.)

Response: Thanks for the comments. As the reviewer said, the default of "minimum % identity" for the software Plasmidfinder 2.1 is 95%. However, under this threshold, we could not obtain the incompatible group of pNDM4-191773, so we adjusted the threshold to "minimum 90% identity",

then we confirmed the incompatibility group of pNDM4-191773 was IncFII(Yp) (90.84% identity). According to the report “In silico detection and Typing of Plasmids using PlasmidFinder and Plasmid Multilocus Sequence Typing, DOI:10.1128/AAC.02412-14”.an identity threshold of 80% was used to detect plasmid replicons in bacterial whole-genome sequencing data.

Line 212: justification for identity cutoff

Response: Thanks for the comments. We referred to the reported studies, which used “>95%” to represent homologous fragments when comparing the similarity of gene sequences on plasmids (DOI:10.3389/fmicb.2020.576823). We suspected that this might be similar to using ANI > 95% to represent the same species when analyzing microbial species. We have used Blastn to verify the identity between sequences and found that the average nucleotide identity is > 99%. We have modified the cutoff in Fig 3, Fig 4b, and Fig S3.

Lines 259-260: explicitly state in the text the significant differences wrt. Susceptibility

Response: Thanks for the suggestions. We have made an extensive modification to the Result section, and we feel very sorry that this sentence has been removed from the original manuscript.

Line 290: reference required

Response: Thanks for the warm suggestions. We have rewritten this sentence, and the reference to Korean strains has been added. (Line 243, 349)

Line 292-294: This sentence is not adequately proved in the manuscript

Response: We feel very sorry for the incorrect description. It is really true as the reviewer commented. We could not demonstrate that the *bla*_{NDM-4}-harbored plasmid can be stably present in bacteria in the presence of different antibiotics. Thus, this sentence was deleted in the revised version. Thanks for the insightful comments.

Line 312-313: reference required

Response: Thanks for the comments. The reference has been added in Line 192.

Line 313-315: reference required, what is the relevance of this sentence to the manuscript findings?

Response: We feel very sorry for the poor writing. We initially thought that the differences in virulence factors were due to the evolution of bacteria. However, further analysis revealed that it was not evolution that caused the lack of virulence factors in the CZA-resistant strains. This sentence has been removed. Thanks to the reviewers for pointing out our problems.

There are syntax errors throughout the manuscript. I have detailed a few examples below, but I recommend that a copy editor go over the manuscript to fix other such errors.

Response: We feel very sorry for the poor writing. We will check the revised manuscript carefully.

Line 28: "were co-existed" remove were

Response: thanks for the suggestions. "were" has been removed. (Line 29)

Line 29: "contained" should be containing

Response: thanks for the suggestions. We have rewritten the Abstract, and this sentence was deleted in the revised version.

Line 32: "which likely caused by" should be "which was likely caused by"

Response: thanks for the suggestions. "was" has been added. (Line 32)

Line 38: spell out CRKP fully. (acronym not defined until the main manuscript)

Response: Thanks for the suggestions. We have changed “CRKP” to “carbapenem-resistant *K. pneumoniae*”. (Line 39)

Line 61: sequences play

Response: Thanks for the suggestions. We have changed “sequence” to “sequences” and changed “IS” to “ISs”. (Line 67)

Line 62: belonging

Response: Thanks for the suggestions. We have changed “belong” to “belonging”. (Line 67)

Line 65: define CPS

Response: Thanks for the suggestions. As this sentence is irrelevant to this manuscript, it has been deleted.

Line 70: "become more deeply"?

Response: Thanks for the comments, and we feel very sorry for the confusion caused to the reviewers. We have rewritten the Abstract and removed this sentence.

Line 120: amplified

Response: Thanks for the suggestions. We have changed “amplificated” to “amplified”. (Line 130)

Line 120: primers

Response: Thanks for the suggestions. We have changed “premiers” to “primers”. (Line 130)

Line 123: "11 hour" should be 11 hours

Response: Thanks for the suggestions. We have changed “11 hour” to “11 hours”. (Line 140)

Line 136-137: and P-values <0.05 were considered significant

Response: Thanks for the helpful suggestions. We have rephrased the sentence according to reviewer's suggestions. (Line 159)

Line 147: retained

Response: Thanks for the suggestions. We have changed "retaining" to "retained". (Line 172)

Line 152: considerable similarity

Response: Thanks for the suggestions. Since the average nucleotide identity could not reveal the relatedness among our strains, this sentence has been removed.

Line 184: remove "in size"

Response: Thanks for the suggestions. We have removed "in size". (Line 241)

Line 196: No space between β and lactamase

Response: Thanks for the suggestions. The space between β and lactamase has been removed. (Line 265)

Line 202: suggesting homology extremely

Response: Thanks for the suggestions. We speculated the reviewer was not satisfied with the writing here, so we have changed the sentence to "This suggested that the same plasmid carrying *bla*_{OXA-181} has spread in Europe." (Line 271-272)

Line 221: "resistant" should be resistance

Response: Thanks for the suggestions, and we feel very sorry for our negligence. We have changed "resistant" to "resistance"(Line 219) and checked the spelling of resistance genes in the manuscript.

Line 278-279: grammar for "this" vs "strains" either "this strain" or "these strains"

Response: Thanks for the suggestions. We have modified here and rechecked the grammar of the full text. (Line 333)

The authors are inconsistent with their use of beta spelled out and the Greek character β . E.g. lines 292 and 298

Response: We feel very sorry for our negligence. We have changed “beta” to “ β ” uniformly as the reviewer’s warm suggestions. The sentence in line 292 has been removed because of the inappropriate expression.

Line 295: "surveillance" should be surveilling

Response: Thanks for the suggestions. We have changed "surveillance" to “surveilling” (Line 383)

Line 299: "patient discontinued imipenem" to "patient's imipenem therapy was discontinued" since the decision to discontinue the therapy was presumably made by the patient's medical team

Response: Thanks for the constructive suggestions. We have revised this sentence in Line 355-356.

March 26, 2022

Dr. Yingshun Zhou
Southwest Medical University
Zhongshan
Luzhou
China

Re: Spectrum02624-21R1 (Comparison of two distinct subpopulations of *Klebsiella pneumoniae* ST16 co-occur in a single patient)

Dear Dr. Yingshun Zhou:

Thank you for submitting your manuscript to Microbiology Spectrum. Your revised manuscript has been reviewed by the same experts who commented upon the original work. All reviewers agree that the manuscript is significantly improved. There remain some minor revisions to improve the work. Once these changes are made, I will move the revised manuscript forward for publication.

Link Not Available

Sincerely,

Karen Carroll

Journals Department
Reviewer comments:

Reviewer #2 (Comments for the Author):

The authors have addressed most of the questions raised. Thanks. Yet, the manuscript still has some mistakes that would require revision. Please see below my questions:

line 63, pathogenic bacteria "have evolved", remove "need to"

line 64, revise "As one of the selective pressures, the antibiotic treatment promotes" to "The selective pressure exerted by the

antibiotics has driven"

line 114, Was 60% coverage criteria too relax? Did you recover partial genes?

line 122, add a reference for kSNP

line 179, change "was assigned to" to "clustered with"

line 189, add a reference for Kleborate

line 205, there was no integer in the square??

line 237, "in successfully establishing drug susceptibility differences" is not a good expression. need to be rephrased.

line 243, "sequence identity 99.99%"

line 245,259 better rephrased as " genetic organization of region surrounding the blaNDM-4..."

line 266, change "need" to "required"

line 336, 338, better change to K. pneumoniae ST16

line 351, perhaps the patient in S. Korea had traveled to China?

line 362, the statement on compensatory mutation was out of place here? Did you detect SNPs in certain genes that may be considered as compensatory?

Reviewer #3 (Comments for the Author):

The authors have satisfied my minor and major comments and I do not have new ones to add.

Reviewer #6 (Comments for the Author):

The authors have addressed many concerns and should be commended on their great efforts towards this revision. The introduction has greatly improved with the removal of extraneous information and the addition of more relevant background. The discussion is also greatly improved and does a good job of putting the results into context. Overall, the manuscript is far more readable.

I have only minor revisions (detailed below) for the current version.

Minor revisions:

Most of my concerns here regard the framing/discussion of evolution in the text. Evolution is not an intentional process and some of the wording in the manuscript makes it sound like an active and intentional effort on the part of bacteria.

Revise line 63: "evolve themselves to meet" is too active. Consider "Pathogenic bacteria face challenging new selection pressures when infecting a new host."

Revise line 65: "After developing antibiotic resistance, parts of bacteria can evolve into" consider "After developing antibiotic resistance, bacteria may persist and play a key role in..."

Line 69: consider using "bacteria" instead of "host" here to avoid confusion because host is used previously to refer to the human.

Line 351: "Unfortunately" should be removed. My interpretation of the data is that the South Korean sequence type is likely more widespread than currently reported in the literature.

Staff Comments:

Preparing Revision Guidelines

- Point-by-point responses to the issues raised by the reviewers in a file named "Response to Reviewers," NOT IN YOUR COVER LETTER.
- Upload a compare copy of the manuscript (without figures) as a "Marked-Up Manuscript" file.
- Each figure must be uploaded as a separate file, and any multipanel figures must be assembled into one file.

- Manuscript: A .DOC version of the revised manuscript
- Figures: Editable, high-resolution, individual figure files are required at revision, TIFF or EPS files are preferred

Please return the manuscript within 60 days; if you cannot complete the modification within this time period, please contact me. If you do not wish to modify the manuscript and prefer to submit it to another journal, please notify me of your decision immediately so that the manuscript may be formally withdrawn from consideration by Microbiology Spectrum.

Manuscript Revisions

The authors have addressed many of my concerns and should be commended on their great efforts towards this revision. The introduction has greatly improved with the removal of extraneous information and the addition of more relevant background. The discussion is also greatly improved and does a good job of putting the results into context. Overall, the manuscript is far more readable.

I have only minor revisions (detailed below) for the current version.

Minor revisions:

Most of my concerns here regard the framing/discussion of evolution in the text. Evolution is not an intentional process and some of the wording in the manuscript makes it sound like an active and intentional effort on the part of bacteria.

Revise line 63: “evolve themselves to meet” is too active. Consider “Pathogenic bacteria face challenging new selection pressures when infecting a new host.”

Revise line 65: “After developing antibiotic resistance, parts of bacteria can evolve into” consider “After developing antibiotic resistance, bacteria may persist and play a key role in...”

Line 69: consider using “bacteria” instead of “host” here to avoid confusion because host is used previously to refer to the human.

Line 351: “Unfortunately” should be removed. My interpretation of the data is that the South Korean sequence type is likely more widespread than currently reported in the literature.

Dear Dr. Karen Carroll

Thank you very much for your letter and the reviewers' comments on our manuscript entitled "Comparison of two distinct subpopulations of *Klebsiella pneumoniae* ST16 co-occur in a single patient" (ID: Spectrum02624-21R1). According to the editor and reviewers' comments, we have made extensive modifications to our manuscript in the previous version, and thanks for the recognition of our work. We have studied the reviewers' latest and valuable comments carefully, which are very helpful for revising and improving our paper. The main corrections are in the manuscript marked in yellow and the response to the reviewers' comments are as follows in bold.

Kind regards.

Yingshun Zhou, Southwest Medical University

Email: yingshunzhou@swmu.edu.cn

Reviewer comments:

Reviewer #2 (Comments for the Author):

The authors have addressed most of the questions raised. Thanks. Yet, the manuscript still has some mistakes that would require revision. Please see below my questions:

Response: In the last revised version, we have supplemented with detailed data as suggested by the reviewer to make our results convincing. Thank you very much for those constructive comments. At the same time, we feel very sorry for the mistakes that still exist in the manuscript and very grateful to the reviewer's professional review work on our article.

line 63, pathogenic bacteria "have evolved", remove "need to"

Response: Thanks for the helpful suggestion. Since the original sentence described bacterial

evolution as an intentional process, which may not be consistent with natural selection. Therefore, we have rewritten the sentence. (Line 63)

line 64, revise "As one of the selective pressures, the antibiotic treatment promotes" to "The selective pressure exerted by the antibiotics has driven"

Response: Thanks for the insightful suggestions. We have rephrased this sentence according to the reviewer's suggestions. (Line 63-64)

line 114, Was 60% coverage criteria too relax? Did you recover partial genes?

Response: Thanks for the comments. When using the CGE server to analyze plasmid replicon types, plasmid MLST, and resistance genes, the identity and coverage thresholds are only required in detecting the plasmid replicon types. According to the previous report (DOI: 10.1128/AAC.02412-14), for a hit to be reported, it has to cover at least 60% of the length of the replicon sequence in the database. Consequently, we had chosen the default parameter settings. For the problems raised by the reviewer, we selected 80% coverage to analyze the whole-genome sequencing data and obtained the same results. We have changed the "60% coverage" to "80% coverage" in the revised version. (Line 114, 241)

line 122, add a reference for kSNP

Response: Thanks for the helpful comments. The reference has been added to the revised manuscript. (Line 122)

line 179, change "was assigned to" to "clustered with"

Response: Thanks for the helpful comments. We have modified this sentence according to the reviewer's suggestions. (Line 179)

line 189, add a reference for Kleborate

Response: Thanks for the comments. The reference has been added to the revised manuscript.

(Line 189)

line 205, there was no integer in the square??

Response: Based on the analysis of the resistance genes of the reference sequence QS17-0029 (accession no. CP024038.1), we found that QS17-0029 carried two *bla*_{CTX-M-15}. Accordingly, we marked the white number in the black square. We feel very sorry for the unsharp number in this picture, which caused confusion to the reviewer. We have enlarged and bolded the font to make it easier for reading. (Fig 2)

line 237, "in successfully establishing drug susceptibility differences" is not a good expression. need to be rephrased.

Response: Thanks for the helpful comments, and we feel very sorry for our poor writing. The sentence has been rewritten in the revised version. (Line 237-238)

line 243, "sequence identity 99.99%"

Response: Thanks for the helpful suggestions. We have modified this part, and other parts with similar descriptions have been modified. (Line 222, 243-244, 282)

line 245,259 better rephrased as " genetic organization of region surrounding the blaNDM-4..."

Response: Thanks for the warm suggestions. We have corrected it and we are also grateful for your point out. (Line 245, 259)

line 266, change "need" to "required"

Response: Thanks for the correction. We have changed the "need" to "required", according to the reviewer's suggestions. (Line 266)

line 336, 338, better change to *K. pneumoniae* ST16

Response: Thanks for the warm suggestions. We have changed all the "ST16 *K. pneumoniae*" to "*K. pneumoniae* ST16" in the manuscript. (Line 336, 338)

line 351, perhaps the patient in S. Korea had traveled to China?

Response: Thanks for the insightful comments. We have reviewed the clinical description of the Korean strain 18-2374 (accession no. NZ_CP041927.1), which was isolated from an 81-year-old male patient with colon cancer. There is no mention in the article that the patient had a history of Chinese travel. However, the authors mentioned that NDM-4- and OXA-181-positive *K. pneumoniae* may have spread within the hospital, so we speculated that the mobile population carrying this strain led to the detection of nearly identical strains in different areas. In addition, this strain may have been widespread, although the relevant reports were rare. (Line 340-341)

line 362, the statement on compensatory mutation was out of place here? Did you detect SNPs in certain genes that may be considered as compensatory?

Response: Thanks for the comments. It has been reported that the fitness cost associated with plasmid carriage can be counterbalanced by acquiring compensatory mutations, either in the plasmid, in the bacterial chromosome, or in both. According to the previous reports, deletion of either *gacA* (*uvrY* in *K. pneumoniae*) or *gacS* in *Pseudomonas fluorescens* was sufficient to completely ameliorate the cost of plasmid carriage (DOI: 10.1016/j.cub.2015.06.024), while in *E. coli*, *sspA* and *oxyR* genes were associated with the compensatory changes (DOI:10.3390/antibiotics10010090). In addition, *rpoS*, *mrgB*, *phoPQ*, *rssB* were related to evolutionary response of *E. coli* under antibiotic selection (DOI:10.7554/eLife.70931). Based on the analysis of a *K. pneumoniae* ST258 strain that colonized a patient over 4.5 years, the evolutions were associated with respiration (*sucC*, *fnr*, *narL*), cell envelope (*pfeA*), capsular

polysaccharides (*epsJ*, *wcaJ*, *rscC*) (DOI: 10.1093/cid/ciy293). In our study, none of the above mentioned mutations were detected. Nevertheless, one of the mutations occurred in chromosome of K191773, is the truncation of *pqiB* gene by *ISAeme19*. PqiB belongs to the mammalian cell entry (MCE) proteins. It has been reported that the deletion of *pqiB* may result in OM defects, and the mutations of *pqiB* were related to polymyxin resistance with *mlaDF*. Whether the truncation of this gene is involved in alleviating the cost of plasmid carriage deserves further investigation. Apart from this, we did not find any SNPs which may be associated with compensatory mutations. What's more, the mutations in the replication initiation genes also contribute to the adaption of resistance plasmid. We compared the sequence identity of plasmid replicon in K191724, K191725, and K191773 and did not find any mutations. Notably, some studies of compensatory evolution do report large-scale plasmid deletions (DOI: 10.1093/molbev/msw163). Therefore, we speculate that IS26-mediated fragment loss in our study is the result of compensatory evolution. We feel very sorry for our poor description which confused the reviewer. We have modified the sentence in the manuscript. (Line 350-352).

Reviewer #3 (Comments for the Author):

The authors have satisfied my minor and major comments and I do not have new ones to add.

Response: Thank the reviewer for being satisfied with our revised manuscript. All of the comments pointed out by the reviewer were significant for this manuscript and helped us to improve the quality of our article.

Reviewer #6 (Comments for the Author):

The authors have addressed many concerns and should be commended on their great efforts towards this revision. The introduction has greatly improved with the removal of extraneous information and the addition of more relevant background. The discussion is also greatly improved and does a good job of putting the results into context. Overall, the manuscript is far more readable.

Response: We feel great thanks for the reviewer's professional review work on our article. As the reviewer said, we have made extensive corrections in the previous version. It was the specific and rigorous writing guidance provided by the reviewer that helped us to improve the quality of this manuscript. Based on these comments, we have carefully modified the manuscript. Thanks again for these constructive suggestions.

I have only minor revisions (detailed below) for the current version.

Minor revisions:

Most of my concerns here regard the framing/discussion of evolution in the text. Evolution is not an intentional process and some of the wording in the manuscript makes it sound like an active and intentional effort on the part of bacteria.

Response: Thanks for the insightful comments. Regarding the mutation and evolution of plasmids, we reviewed relevant literature. Plasmid evolution is dependent on the fitness cost and recent studies have identified resistance genes carried on plasmids are a major source of plasmid fitness costs. At the same time, it has been reported that plasmid stability is readily improved by the IS26-mediated deletion of costly regions from the plasmid backbone. Therefore, we hypothesized that the loss of IS26-mediated fragments containing resistance genes that occurred in our study was intended to reduce the fitness cost. Mutations represent a fitness advantage,

which enables the adaptive variants to be more competitive. As the reviewer said, mutation and evolution are both natural processes that help dominant bacteria to be selected. Therefore, we emphasize longitudinal isolation of clinical specimens to investigate the process of bacterial evolution. We feel very sorry for being too subjective in describing the plasmid evolution, and the corresponding sentences have been corrected as the reviewer suggested.

Revise line 63: "evolve themselves to meet" is too active. Consider "Pathogenic bacteria face challenging new selection pressures when infecting a new host."

Response: Thanks for the warm suggestions. We have revised the sentence according to the reviewers' suggestions. (Line 63)

Revise line 65: "After developing antibiotic resistance, parts of bacteria can evolve into" consider "After developing antibiotic resistance, bacteria may persist and play a key role in..."

Response: Thanks for the warm suggestions, which were extremely helpful for our manuscript. We have revised the sentence according to the reviewers' suggestions. (Line 65-66)

Line 69: consider using "bacteria" instead of "host" here to avoid confusion because host is used previously to refer to the human.

Response: Thanks for the careful checks. We are sorry for our carelessness, and we have corrected the "hosts" to "bacteria". (Line 69)

Line 351: "Unfortunately" should be removed. My interpretation of the data is that the South Korean sequence type is likely more widespread than currently reported in the literature.

Response: Thanks for the helpful suggestions. It was true as the reviewer said that although the patients in these two studies did not have overlapping travel histories, it cannot be ruled out that other patients or caregivers who also carried this strain acted as the source of transmission.

Therefore, attention should be paid to the global spread of *K. pneumoniae* ST16. We have modified the manuscript according to the reviewer's suggestions. (Line 351-353)

March 31, 2022

Dr. Yingshun Zhou
Southwest Medical University
Zhongshan
Luzhou
China

Re: Spectrum02624-21R2 (Comparison of two distinct subpopulations of *Klebsiella pneumoniae* ST16 co-occur in a single patient)

Dear Dr. Yingshun Zhou:

Your manuscript has been accepted, and I am forwarding it to the ASM Journals Department for publication. You will be notified when your proofs are ready to be viewed.

Sincerely,

Karen Carroll
Editor, Microbiology Spectrum

Journals Department
Supplemental Material (revised version2): Accept